# Web Page Design Recommendations for People with Down Syndrome Based on Users’ Experiences

**DOI:** 10.3390/s18114047

**Published:** 2018-11-20

**Authors:** Lucía Alonso-Virgós, Luís Rodríguez Baena, Jordán Pascual Espada, Rubén González Crespo

**Affiliations:** 1School of Engineering and Technology, Universidad Internacional de La Rioja, Avda. de la Paz, 137, 26006 Logroño, La Rioja, Spain; lucia.alonso.virgos@unir.net (L.A.-V.); luis.rodriguez@unir.net (L.R.B.); 2Department of Information Technology, Universidad de Oviedo, Calle Valdés Salas 11, 33007 Oviedo, Spain; jordansoy@gmail.com

**Keywords:** website accessibility, website adaptation, Web Content Accessibility Guidelines 2.0 or WCAG 2.0, Web Content Accessibility Guidelines 2.1 or WCAG 2.1, Down syndrome

## Abstract

At present, there is a high number of people with Down syndrome interested and trained to be an active part of society. According to the data extracted by our surveys we know that only 6% of the population with Down syndrome feels isolated in daily activities. However, when the activity requires the use of a computer, the percentage of people who feel isolated increases to 18%. This means that there are obvious website accessibility barriers that make it difficult for users with Down syndrome. To solve this problem, it is considered necessary to make an exhaustive study about Down syndrome. We know that the trisomy of chromosome 21 causes a series of symptoms that directly affect ones Internet browsing capabilities. For example, speech disturbances make communication and speed difficult. This guide is based on a neurological study of Down syndrome. Alterations in listening make understanding audio, retention of audio concepts and speed difficult. The alterations in the physiognomy of movement make it difficult for them to act quickly. Many of these alterations are caused by cognitive disability. After assessing the needs, the benefits of Web Content Accessibility Guidelines 2.0 (WCAG 2.0), and the existing usability guidelines are analyzed and those that may be useful for this profile are extracted. User tests are carried out through two websites developed specifically for this study with the aim of demonstrating the level of effectiveness of each of the planned guidelines. Considering the neurological characteristics of this intellectual disability, research is developed that seeks to extract a list of useful accessibility and usability guidelines for web developers.

## 1. Introduction

Down syndrome was considered a serious intellectual disability 30 years ago, but in recent years, several programs have helped to change the classification of Down syndrome to a medium-level intellectual disability [1]. The most important programs that have made this disability go from being considered a severe intellectual disability to medium-level intellectual disability are early care programs, educational inclusion programs and technological advances.

In the field of website accessibility there are no studies on the evaluation of such guidelines. For this reason, the purpose of this research is on the one hand to draw conclusions about the opinion of the users with Down syndrome to know what their most pressing needs are, and, on the other hand, to test all those accessibility and usability guidelines to check if they are indeed useful for this user profile. The objective is to conclude with a series of useful recommendations for web developers interested in website access for people with Down syndrome. To achieve this objective, two studies were carried out:(1)A study whose purpose was to know the real needs of users with Down syndrome. To this end, surveys were prepared and sent to 87 Spanish associations related to Down syndrome. These surveys were answered by the users with Down syndrome, there were a total of 112 volunteer participants.(2)User tests to verify the results of the surveys. These tests were specially designed for the study.

In this article we define the importance of website accessibility for the inclusion of people with Down syndrome. We defined the existing accessibility and web usability guidelines and analyzed the newly approved Web Content Accessibility Guidelines 2.1 guidelines. We present our intervention, which consists of carrying out a survey and different user tests. The survey aims to find the needs that the users with Down syndrome believe themselves to have. The tests are intended to demonstrate the effectiveness of existing guidelines.

We present the trials that were conducted during the research, including those trials that were modified during the tests to better fit the search for the result. We offer the results of each test and a series of useful recommendations for web developers. In the conclusions we make an analysis on the importance of working this method in ​​web accessibility and usability.

The concepts of accessibility and usability are analyzed first in this paper. Especially the advantages offered by web accessibility and WCAG to people with cognitive disabilities. As our goal is to direct the research towards a specific user profile, surveys are conducted and sent to 87 Spanish Down syndrome-related associations. There were 112 volunteers with Down syndrome interested in participating. The methodology of the surveys is explained in this paper in terms of its design and its resolution. Many of the results of the surveys are published in this paper. The purpose of these results is to be able to design some tests that prove a series of web design recommendations. So, these results and the references of author and WCAG are considered for the approach of the recommendations and for the design of the tests. There are 45 essays distributed in 14 tests, web tests or eye tracking tests. The methodology of all of them is detailed in this paper. As well as also the most relevant results obtained. Finally, interesting conclusions are drawn. These conclusions represent useful recommendations for developers or web designers.

## 2. Background

Within the field of website accessibility, there are several recommendations, ordered according to their purpose. In this research, we will analyze those that benefit users with Down syndrome, according to the results of the surveys we conducted and according to the previous and exhaustive study of their profile. After completing the research, new Web Accessibility Initiative (WAI) guidelines appeared, which from now on we will call WAI, many of them aimed at people with cognitive disability. Although they have not yet been formally approved, we analyze them based on the results of our work in this paper. Although they had not been formally approved when we did our tests, we analyze them based on the results of our work in this paper.

### 2.1. Usability and Accessibility

Technology plays an important role in the lives of people with disabilities [2]. In web development, we have a series of universal procedures divided into accessibility and usability. The purpose of website accessibility is to ensure that a website can be used by as many people as possible, regardless of their knowledge or skills or of the technical characteristics of the equipment to be used. This is done, thanks to four principles: Perceivable, Operable, Understandable and Robust. Each of these principles has a series of guidelines that describe how to apply each of the guideline [3]. In the field of accessibility, there are also guidelines to people with cognitive disabilities [4], however, there are no published studies on the evaluation of such guidelines. We cannot guarantee that the guidelines are effective or how effective they are.

The purpose of usability is to ensure that the use of a website is practical, simple and effective. The navigation is intended to be effective, efficient and satisfactory for the user [5,6]. Considering the appropriate usability guidelines can reduce the risk of errors during navigation and increase the ease of use, which not only benefits the user but also their objectives and context [7,8]. In the field of usability, we find a study that tests different usability guidelines for mobile devices in people with Down syndrome. Although our work is not focused on mobile devices, we find this study interesting, since there are very few works aimed at the technological inclusion of people with Down syndrome. The tests are performed by people with cognitive disabilities and determined that the most effective navigation method is “drag” [9].

### 2.2. Website Accessibility

There are guidelines on website accessibility and usability that aim to solve all or almost all the needs that different disabilities or difficulties demand. Regarding Down syndrome, the importance of getting the computer to this user profile in other occasions has been demonstrated [10]. However, at least until now, there are few works on website accessibility aimed specifically at users with Down syndrome. The interest of this user profile is its complexity. Syndrome means “set of symptoms”. All individuals differ from each other and not all have to present the same degree of disability in each symptom. However, the sum of several symptoms can be an inclusion handicap, especially the one concerning cognitive disability.

Almost all existing research related to Down syndrome assesses their physical and non-cognitive difficulties. For example, in the field of mobility, it is known that for users with Down syndrome it is easier to use a mouse than a keyboard, since they show difficulties in entering text using keyboards [10]. This statement was demonstrated through surveys of users aged 4–21 years. However, it would be necessary to test this hypothesis. There are also a multitude of surveys aimed at finding the existing difficulties, but these surveys may have limitations, since they are aimed at very diverse users and usually begin with people of 4 years of age. For example, people under 6 years old may not have learned to write and it is natural for them to feel more comfortable with the mouse than with the keyboard [10].

To date, there has not been an analysis of the real abilities of Down syndrome patients, but there are ethnographic studies that show that users with Down syndrome have sufficient skills to perform basic tasks on computers, such as text processing or data entry [11]. There are also works related to the advantages of the computer in education, communication or entertainment activities [12]. It is interesting to know how computers can help the educational development of students with Down syndrome and how they can help their professional development and independence. There is a study that collects the navigational difficulties of 600 children with Down syndrome and raises the related design challenges [13]. These studies show the possibility of inserting people with Down syndrome into work activities, even those that require the use of new technologies.

#### Website Accessibility for Cognitive Disability

We analyze that in the field of website accessibility it is possible to distinguish the need for a cognitively accessible website. However there has been little work to date in this area. WAI recommendations related to cognitive disability present evaluation difficulties because they are general recommendation [14]. Currently, it is intended that WCAG improve the guidelines, including guideline specifically for cognitive disability. In June 2018, a new WCAG recommendation was approved. The WCAG 2.1 includes guideline specially oriented to cognitive disability. There are 17 novel guidelines. Of the 17 new guidelines of conformity (success criteria) there are those that are more related to cognitive disability: Criterion 1.3.5 “Identify Input Purpose”, which aims to offer content in different formats, for example, with a simpler composition. Criterion 1.3.6 “Identify purpose”, aims to determine by code the purpose of the user interface components, icons and regions. This can be useful for people with memory problems. Criterion 1.4.10 “Reflow”, aims to facilitate the content by avoiding the scroll bar. The criterion 1.4.11 “Non-text Contrast” aims that all the content (including graphical objects) have a contrast of 3:1. The criterion 1.4.12 “Text Spacing” aims that the textual content is offered visually in an appropriate way. Criterion 1.4.13 “Content on Hover or Focus” which aims that the hidden content is not lost. Criterion 2.2.6 “Timeouts” aims to preserve the privacy of minors when there is inactivity and the data is retained. Criterion 2.5.1 “Point Gestures” intends that all functionality that uses multipoint gestures, can be used with a single pointer gesture. Criterion 2.5.2 “Pointer Cancellation” relative to the “singer pointer” functionality. Criterion 2.3.3 “Animation from Interactions”, which aims to disable the animations that are triggered by some user interaction. This can be useful for people with attention problems. Criterion 2.5.3 “Label in Name” intends that visible and non-visible labels have consistency. Criterion 2.5.4 “Motion Actuation” which aims to include alternative methods of interaction to the movements of the device (to shake the mobile) or the movements of the user detected by a camera. This can be beneficial for people with spasms, or with cerebral palsy. Criterion 4.1.3 “Status Messages” also refers to screen readers.

Although the intention is to improve the field of cognitive accessibility, it is emphasized that they do not cover all the needs of these users. To date, no evaluations have been published on web guidelines for people with cognitive disabilities. There are no specific developmental guidelines that have been tested with users with Down syndrome [14]. For this reason, this research aims to address this problem and quantitatively evaluate the WAI and existing guidelines on usability aimed at cognitive disability to determine if they are indeed useful. In addition to evaluating existing guidelines, the research concludes with a list of useful website accessibility and usability recommendations aimed at developers who want to create cognitively accessible web pages.

## 3. Research Development

This study presents two complementary studies: a survey that aims to find out the real needs of users with Down syndrome and two types of user tests that aim to extract a list of useful guidelines for access and use. The purpose of the survey is to find the needs that the users themselves believe they have. With the results obtained in the surveys and with the previous analysis of the neurological profile of Down syndrome, it is determined which guideline of the existing ones can be effective.

The user tests intend to determine if these guidelines are effective. Alternatives are proposed to many of the guideline to fit the real needs of the participants. The aim is to conclude with a series of useful recommendations for web developers.

### 3.1. Survey

It is difficult to learn new words and to understand and express oneself through linguistics for people with Down syndrome. The most common problems during reading are the confusion of phonemes, the replacement of words with similar ones and the inversion in the order of letters or syllables. Also, there are typographies that can be confusing because they prioritize their aesthetic character to the functional character. For example, the Gill Sans font offers letters different with the glyph so similar that they do not differ from each other. In this typeface, an “l”, an “i” and the number “1” with the font type Gill Sans offers identical characters, I-I-I. So, that the volunteers understood the text was a priority for us. Many of the recommendations collected for the design of the survey were related to typography, grammar and format [15].

The survey design complies with the accessibility recommendations on format and writing. For this to be fulfilled, and before sending the surveys to the associations, a test survey was designed. The test survey was sent to three associations and the recommendations offered by the expert staff were recorded. When the three centers approved the format and the writing, it was sent to the 87 associations planned.

#### 3.1.1. Survey Design Methodology 

The design and structure of this survey is the result of work and consultations with a lot of associations. A series of published guidelines were obtained for the preparation of the survey form too. The purpose of these guidelines is to improve the format and content [16]. Our main difficulty was the use of appropriate language and wording understandable by the interested users, so we also sought advice from specialized personnel. For example, they told us what kind of questions more easily answered and what number of answers are appropriate. It was preferred to use the same communication scheme used by the associations to avoid disorienting users when accessing information and answering forms. The review by the staff of the different associations has been very useful.

The tests were in paper format, not online. The first survey design was reviewed by an expert in the creation of teaching materials. Since the survey was part of the investigation and we were not sure if the theoretical recommendations would be useful, trial and error tests were made with several participants. This helped to adjust the writing, until the questions could be answered autonomously by a specific profile. There were 25 participants who fulfilled the study profile. They participated only in the survey tests, not in the web tests or in eye tracking tests. A series of new recommendations were collected. The surveys were modified and considered valid. With the new survey format all participants responded without any use problems:Text was left justified, and not centered, to avoid confusing the users with the space between words.White background and dark gray letters were used, instead of black, to prevent the text from causing blurring. Color was only used on the cover, as a persuasive method. Do not forget that participation is voluntary.The questions were written using secondary language, avoiding technicalities, complex words, or redundancies. Double meaning is not used either, the sentences are literal and short (less than 15 words).Serif typography is used, avoiding italics and using the bold font to highlight the text. The typeface used is Sans Serif.

The survey is divided into four different sections that motivate its completion.
A form that collects data to establish the sample characteristics (sex, age, level of studies, etc.). Personal information such as name, city or school was not collected in any of the cases. The field is by selection, so the respondent must select one of the proposed answers.A form that assesses the difficulty level of different computer activities based on four variables (never, sometimes, many times or always). The field is by selection, so the respondent must select one of the proposed answers. The results of these types of questions are useful to find out if users have obvious access difficulties and if they are aware of them. It must be remembered that the surveys are answered by the interested parties, so that in these reports it is expected to find problems identified by the users.Open space that collects the most navigated portals. The field is free form.Alternative space to provide complementary data. The field is free form. This field was not obligatory and nevertheless 48% of the respondents answered it voluntarily.

All questions offer a section of alternative answers so that users can respond differently to the questionnaire. For example, a free copy space is included in the fields in which the respondent must select one of the proposed options. In this way the respondent can respond either by selecting the option that interests them, or by writing their own answers:Multiple choice type answers, with a maximum of six possible answers.Various shipping formats, paper and digital format with auto response boxes were offered.

#### 3.1.2. Methodology to Increase the Participation in the Survey Results

The surveys are sent to 87 Spanish Down syndrome-related associations and was answered by 112 voluntary users (67% of the eligible candidates contacted). It is important to emphasize that to participate not only must the user agree, but also his/her family. Both the tests and the survey required proof of project approval signed by the families.

There was no age limit, however there was a greater number of participants between 15–35 years old. The average age of the participants was 17 and the deviation is 26. It is not surprising that this has been the most prominent age range. On the one hand, students with Down syndrome have access to secondary school with several levels of curricular lag. This means that they enter secondary school with at least 15 years old. In secondary school, TIC competence is compulsory, so it is logical that those are students are interested in computers. On the other hand, the number of students enrolled in advanced courses, such as vocational training, has increased considerably since 2008. The insertion in the labor market demands that workers have skills in new technologies (telephones, PDAs, computers). For this reason, it is understandable that there is a high number of people under 35 years of age was interested in participating in the survey.

It is important that the users be able to respond to the survey without any help. However, there was always an assistant. The assistant should only clarify doubts about the questions or serve as a guide as to how to respond. The assistant cannot give the answers. Participants did not receive any help to collect information related to their writing, expressions, etc., directly from them.

#### 3.1.3. Survey Results

Table 1, Table 2, Table 3 and Table 4 shows the statistics of the participants’ data. The surveys show that 3.5% of people with Down syndrome use the Internet daily, 40% of whom need to use the Internet at school or at work. This is a significant fact, considering that the figure does not depart much from the total percentage of people with Down syndrome between 15–35 years old, 78.57%. It could be concluded therefore that more than 50% of people with Down syndrome in the 15–35 years old age group need to use the Internet daily. We know that the trisomy of chromosome 21 causes a series of symptoms that directly affect ones Internet browsing capabilities [17].

Figure 1 shows the statistics of the day-to-day activity questions. That is, questions that a priori are not related to the Internet. Many of these answers served as motivation for the study. For example, of the total number of volunteers, there are 17% of users who suffer navigation problems daily and 5% of people who cannot use the Internet due to access problems. There is 18% of people with Down syndrome who feel discriminated against when they participate in activities that require the use of a computer. This figure is important, since it differs greatly from the percentage of people who usually feel discriminated against because of their disability (6%, Figure 2 and Figure 3). That is, there is a 12% of people who do not feel discriminated against on a regular basis but feel discriminated when the activities are related to the computer [16].

Figure 2 shows the statistics of the questions about Internet use. It should be noted that the answers were given by the participants themselves. This means that they are not objective answers, but rather personal appreciations about their abilities.

In the web tests we were able to obtain more objective statistics about competencies, based on actual results, not assumptions. Some users mention the technology they used. Only 1% use adaptive tools, such as accessible mice or keyboards.

Thanks to these results we adjusted the tests. For example, in color tests the screen configuration responses gave us the idea of ​​including an additional web test. This test consisted of an adaptable menu for simple changes of the webpage appearance, for example, the color of the background, the color of the buttons, etc.

The most visited websites were YouTube, administration websites, medical services websites. The most used search engine is Google. In this paper, percentages are not included on the most used website because the field of answers was free form. These data serve as motivation for future tests.

The results of the surveys were compared with the recommendations proposed by the WCAI and useful conclusions were drawn for the design of the next test.

The percentage of respondents who work or had worked was 46%. In the autonomous communities of Spain there are a series of job options oriented to the workforce insertion of people with Down Syndrome. These jobs include baker’s assistant, janitor assistant, operator, sales assistant, service assistant, telephone operator, etc. Many of these professions require the use of Personal Digital Assistants (PDAs) and other technological tools. According to the surveys, 23.2% of the users experienced daily problems to learn how to use technological tools. In addition, 48.2% of respondents found it difficult to find tools or buttons without help or supervision.

From the results of the surveys it is concluded that only 15.2% of users need help almost always to maintain or change body posture. Although it mainly refers to users older than 55 years, this data is useful with the eye tracking test. Just in case, for this test we have a help person.

From the results of the surveys it is concluded that 51.8% of the users have problems with the keyboard and that 26% usually have problems with the mouse. This problem may be due to a cognitive deficit, rather than a motor deficit. These data are useful for the development of the tests. For example, in the text link test, the working time is measured using only the mouse. In all web browsing tests, the use of the keyboard is avoided, except in those in which the use of the keyboard is tested.

According to the surveys, 69.6% of respondents need more time to perform tasks related to the Internet, of which 42% require more time in all or almost all their connections. These data are useful for visible stopwatch tests.

According to the surveys, 51.1% of the respondents presented difficulties during the reading, 23.2% said that they had difficulties always or almost always. Approximately 55.3% find it difficult to express themselves through abstract language. These data are useful in the iconography tests.

According to the results, 74.1% of respondents said they make mistakes while performing tasks through the computer. There was a 25% who made mistakes always or almost always. Providing information about how to correct errors allows people with Down Syndrome to fill out forms and identify and correct their mistakes while browsing.

According to surveys, 9% of users have problems with the colors of the web. This has been considered for background and text color tests, and for button color tests.

Although only 11% believe they present common problems reproducing a multimedia element, it was considered useful to perform tests to verify this claim. There is a 20% that manifests having problems with reading a web page. Typography tests were therefore carried out.

### 3.2. Tests

The access barriers suffered by people with Down syndrome are related to a series of physical and neurological conditions caused by the total or partial copy of chromosome 21 [18]. This research proposes a series of user tests that aim to extract a list of access and use recommendations [16,19] useful for web developers.

There were two types of tests:

Web tests. These tests consisted in presenting users Down syndrome with two websites, one inaccessible and one accessible, considering usability and accessibility guidelines. The inaccessible website was intentionally inaccessible to check the time lost that is invested when certain guidelines are not considered. Several tests were carried out, all of them related to the navigation activities most demanded by this user profile. It should be noted that the navigation preferences of this user profile are the same as those of user profiles without intellectual disability [18]. Time comparisons were made between the two websites, the accessible website and the inaccessible website. These tests were performed with 25 volunteers with Down syndrome aged between 18–35 years old and with interest in new technologies.

Eye tracking test. With these tests it was possible to measure the movement of the retina of the participants, in this way information about the attention can be obtained. People with Down syndrome usually have attention deficit [20]. With these tests we try to find out which are the most useful web design and development strategies to fix the attention of a user with Down syndrome. These tests were aimed at 25 volunteers with Down syndrome between 11–25 years old and with an interest in new technologies. In these tests, a limit was established in the age of participants. The average age of the participants was 18. The deviation is 4.2 years old (Table 5). In this table the ages of the test participants are presented. These participants were filtered by our requirements, so information about their age is important. They are people who attend secondary education centers, have access to new technologies and an IQ suitable for inclusion centers. There are also people integrated into jobs, who use new technologies and who have adequate skills to carry out some activities. All these participants were part of both the user tests and eye tracking tests. For the objectivity of the tests, it is important that none of the participant have additional symptoms, aside from Down Syndrome and that none of the volunteers have diagnosed ADHD.

The results of the tests must seem reliable to the public for whom the results are intended. Volunteers with a range of IQ between 50–70 were selected for the tests. A mild or moderate IQ will facilitate the conclusion of the tests. In addition, all volunteers must have a relationship with computers, for example, in their school or workplace. None of the participating users was new to the Internet. All those tested uses the Internet at least three times a week. This makes the resolution of the tests easier and it also makes the results more objective in terms of the need for website accessibility.

#### 3.2.1. Web Test Methodology

Before the tests, prototypes of how the user tests will be carried out were developed. To do this, certain requirements are extracted from the conclusions presented above and collated with author references. The web tests consist of the design of two websites, one accessible and the other inaccessible according to the recommendations included in the previous conclusions. Both websites have similar tasks to make comparison easier. The two websites have the same structure: one HOME page and five PAGES where the tests are hosted (Figure 3). The user tests have text, audio, images, video with and without subtitles, configuration and a form. The results of the tests are extracted with quantitative data, measuring the reaction times of each user and finding temporary gains in the application of certain guidelines, but qualitative results are also extracted, according to the reactions of the users, their comments and their impressions after the tests. This was very useful to understand the level of frustration.

Before beginning the tests, a script was designed with the dialogues and enough technical indications to be able to carry out the test. There is a script for the two tests, one for the web tests and another for the eye tracking tests. It is important to learn the memory dialogues, since its wording has been specially designed so that a person with Down Syndrome may understand the test without needing help. The script in paper format has a space on both sides to note possible interferences of linguistic comprehension during the tests, so if more than two people with Down Syndrome show a lack of comprehension somewhere in the established script, it must be modified before continuing with the rest of the tests. While the users do the tests, they must indicate their step by step progress, to ensure that they understand the objective of the tests and to consider their speed of reaction and understanding. Their gestural or verbal responses were also annotated.

This test is done individually and in a separate room. The users enter one by one and they are explained the content of the tests according to the script. The same order is not always used. Volunteers completed each of the tests in an arbitrary way, so that they do not learn from the experience and the results are more objective. The methodology and the arbitrariness of the tests vary with each subject, since each presents different needs from the rest. That is, not everyone participates in all the tests, or because their disability does not allow them to understand some of the sections, or because fatigue prevented them from continuing. It is considered that people with Down Syndrome are prone to distraction. The objectivity of the results could be altered if they are distracted by fatigue.

For the performance of each test, users work on a specific page of the website resolving the tasks by their own means and narrating each of their steps in a loud voice. They are offered an assistant during the tests because one of the objectives of the research is that people can develop their own autonomy. The assistant cannot provide clues about the tasks, and only resolves doubts as to the meaning of buttons or phrases that the volunteer may not know. The volunteer is explained the meaning of those buttons and tools that may be more problematic, such as the CONFIGURATION button, before initiating each test. Help is offered to them to solve specific doubts in the understanding of the task during the tests.

The users can use the time they wish to perform each of the tests, up to a maximum of 10 min per attempt. This time is proposed on the basis to the recommendations of the workers of the collaborating associations. According to the recommendations of the experts, if a volunteer takes a long time it is possible that they get tired and lose concentration, so as the time of each test depends on the speed of the user, breaks are offered between the tests so as not to exceed the anticipated maximum time. The tests are recorded on the computer with a screen recording software that provides information regarding the time and duration of each test. However, all tests are manually timed.

#### 3.2.2. Eye Tracking Test Methodology

In addition to the web tests, an eye tracking tool was used to verify that the retina of the users followed the prompts of the contents. One of the most representative physical characteristics of Down’s syndrome is the presence of slanted eyes. It is common to have visual deficit that causes loss of attention. In addition, they present difficulty in spatial representation, visuospatial orientation and planning. The visual deficit is linked to underlying problems related to other areas of cognition [21].

The testing of the eye tracking tests is done individually and in an independent room. The subject user must be seated in front of the computer screen and with the back straight without being able to move the head. This was a challenge for people with Down syndrome, since some had postural problems, so a laptop was used to be able to tilt the screen according to the needs of each user.

For the tests, all screens of the two websites, website A (accessible) and website B (inaccessible) in the software were loaded. The test consisted of each user viewing each screen and responding to a series of location questions, for example: “look for the CONFIGURATION button”. The results provide statistical data of the total number of visual visits that a specific area of each of the screens presented; data on the response time of the retina to locate a specific area; and heat maps of the areas most visited by the retina of each subject.

The user cannot move his head during testing. This is a challenge so tests were simulated before the true testing. The same eye tracking software does a pre-run check that verifies the characteristics of the retinas of each subject during simulation tests. Once the demanded objective is found, the user must say “I found it” out loud. For those subjects who presented communication problems, it was enough for them to make an approval sound indicating that they had found the indicated button, for example, many chose to say “already”. With this it is possible to configure and adjust the eye tracking glasses to each user. The real tests start after the adjustments of the simulation tests. Two types of results are extracted from eye tracking tests: Statistical data, that record the global time that users take to locate a certain part of the website;and heat maps, that registered the areas most reviewed by the users.

#### 3.2.3. Multimedia Element Evaluation

During the perception of multimedia content not only does one need to see and hear the content, but the user also needs to understand it [22]. The frontal lobe of Down syndrome subjects presents a reduced volume, larger microcolumns and fewer cells. This causes a lesser development of linguistic skills and memory, less ability for verbal work, memory problems and difficulties in abstract thinking, which can cause reasoning problems in long or complex sentences, and a low understanding of content with figurative meaning [17]. The Internet is one of the largest communication channels and demands discursive skills [12]. Subtitles that are too long or with more than three sentences can cause the same comprehension problems as complex words.

The parieto-occipital-temporal lobe of people with Down syndrome has a normal volume but presents alterations of the microstructures of the pyramidal cell. This has negative effects on visual and/or facial recognition. For example, during a conversation between several participants, the difficulties in visual recognition and in the differentiation of the interlocutors can affect the understanding of the dialogues [16]. Memory is another skill demanded by the Internet [23,24]. The temporal lobe of people with Down syndrome has a reduced volume, a reduced number of granular cells and an alteration of the microstructure of pyramidal cells [25]. This causes less episodic memory, which can cause problems during the dialogue follow-up.

#### 3.2.4. Multimedia Elements Test

There were six trials on two websites for this test. This test is designed to determine which method of reproduction of multimedia content is most appropriate for people with Down Syndrome. For this, three trials will be carried out with 31 s long videos with the option of looping. The objective is for the user to pause the video when the content indicates it. For example, the narrator may say “pause the video when you see an apple”. This statement can be issued aurally, without a narrator, or verbally with and without a narrator. That is, the message is offered through audio and/or text:Audio + video + narrator. Audio + video. Tests 1–2: This essay consists of playing two files with audio and video. One of them with a visible narrator, and the other without visible narrator (Figure 4 and Figure 5).Audio + subtitles + video + narrator. Audio + subtitles + video. Tests 3–4: This essay consists of playing two files with audio, subtitles and video. One of them with a visible narrator, and the other without visible narrator.Video + subtitles + narrator. Video + subtitles. Tests 5–6: This essay consists of playing two files without audio. One of them with subtitles and narrator visible, and the other with subtitles and no visible narrator.

The tests ask the user to perform a concrete action. It is concluded that the user has understood the test when the action is carried out efficiently. In this case, volunteers are expected to pause the file when they see a specific object.

There were 25% of the users who do not finish the test when there is audio and no narrator. There was a 0% of users who do not finish the test when there is audio and a narrator. It can be concluded that the incorporation of a visible person inhibits the abstraction and helps the understanding of the content. The temporal gain visualizing a narrator is 166 s. The guideline 1.2.1 Audio-only and Video-only (Prerecorded), 1.2.8 Media Alternative (Prerecorded) and 2.2.2 Pause, Stop, Hide. The temporary gain applying these guidelines is 86.1% (Figure 5).

All the test subjects finished the test, navigating Website A and Website B. The incorporation of a narrator, plus subtitles and more audio offers an average temporal gain of 9 s. If compared with the results obtained in the previous test, the temporal difference is significant. When subtitles are added the narrator ceases to be essential for the understanding of the message. Applying the guideline 1.2.1 Audio-only and Video-only (Prerecorded), 1.2.3 Audio Description or Media Alternative (Prerecorded), 1.2.8 Media Alternative (Prerecorded) and 2.2.2 Pause, Stop, Hide there is a temporary gain of 41.6% (Figure 6).

All (100%) of the users finished the tests without audio, with and without a visible narrator. The incorporation of a narrator plus subtitles and without audio offers an average temporal gain of 30 s. Audio is an important element during the inhibition of abstraction in multimedia content, especially when there is a narrator. The reduction of time of the test without audio using visible narrator and subtitles, fulfilling the guideline 1.2.1, 1.2.8, 2.2.2 is 43.7% (Figure 7).

If all the tests are compared to each other, we can put for the several recommendations: 50.2% of the users understand the message more efficiently if there is a narrator, while 25.3% of the users understand the message with more efficiency if there are only subtitles. A 24.9% of users understand the message more efficiently if there are subtitles plus audio. No users understand the message if there is only audio plus a visual image, but without a narrator that provides audio (Figure 8).The presence of a visible narrator disinhibits the abstraction when there is a file with audio. The temporary gain including a visible narrator to an auditory file is 86.1% with respect to the rest of the options.The inclusion of subtitles and audio in parallel can affect the user’s attention. The temporary gain suppressing subtitles is 41.6%.When the audio is deleted, and the subtitles are preserved, the temporary gain is 43.7%.During the test, looping of the multimedia files was allowed. The intention was to verify that a user reacts better to the message when he can pause and restart the video. It is verified that when a message is reproduced half-heartedly, without starting from the beginning, the user’s understanding is 0%.Users show greater security in the face of the written text, especially if it is accompanied by a narrator. When only audio is broadcast many users were left bewildered. There was a 68% that showed frustration before the audio without a narrator and without written text. That frustration is manifested with gestures or sounds of displeasure. Record this percentage during the test was not intended as an objective, but it was considered interesting because there was a high percentage. All (100%) of the users demanded a “pause” button, and 87% restarted the video because they did not understand the message correctly during the first reproduction.

#### 3.2.5. Sound Evaluation

Down syndrome causes hearing impairments in the middle ear, which can cause hearing problems [23]. However, this would not be a problem if there were no problems with attention deficit. The attention deficit is caused by alterations in the cerebellum, frontal lobe and temporal lobe [20,25]. The tests ask the user to perform an action. It is concluded that the user has understood the test when the action is carried out efficiently. In this case, volunteers are expected to go to the “tools” option and change the background color. Since the word “tools” can be an abstract word, a text link that says “here” is used for this test.

#### 3.2.6. Sound Test

There were nine trials on two websites for this test. This test has been designed to determine the efficiency of dB control of auditory messages and background noise, issuing two messages that invite action: one reproduced at 15 dB with a slight background noise, and another at 15 dB without background noise, and the same at 20 dB and 50 dB. On the other hand, two types of file were offered: one with sound adjustment tools, pause options, etc., and another that played automatically. All audios contain messages that ask the user to complete a task. In this way, listening is not only measured in time, but also by understanding the message, Figure 9 and Figure 10. Alterations in listening, cognition and social skills of this profile may affect the use of the Internet. For example, when a user searches for information or needs to memorize necessary concepts while browsing [5].

There was a 16.7% of the test subjects that does not finish the hearing test of 15 dB. None of the subjects did not finish the test with a volume higher than 20 dB. The temporary gain by raising the volume is 118 s. People with Down syndrome have physical alterations that affect the ear and can cause hearing loss. However, according to surveys, only about 1% use a hearing aid. This test does not aim to help people with severe hearing problems, but it does want to help users with mild hearing problems. The reduction of time of the test fulfilling the criterion 1.4.7 Low or No Background Audio and using 20 dB, instead of 15 dB, is 82.73% (Figure 11).

If all the results are compared, some conclusions can be drawn. There is a 37.0% of users that use audio settings. There is a 33.7% of users who do not use the audio settings but respond well to messages of 20–50 dB. There is a 29.39% of users that do not use the audio settings but respond well to messages of 20–50 dB with a slight background noise (Figure 12).When a message is issued between 20 dB and 50 dB there is a temporal gain of 82.7%.Background sounds cause incomprehension of the message. There was a 13% of volunteers with Down syndrome who do not understand a message issued at a normal volume of 20 dB or 50 dB with background noise.Having control over volume and reproduction allows users to be autonomous, and 91% of the volunteers reproduced the audio more than once.

#### 3.2.7. Textual Content Evaluation

People with Down syndrome may have problems during language learning and during communication, due to frontal and temporal lobe involvement. These problems can hinder figurative language, abstract language, non-literal language, and discourse, where the use of episodic memory is required [25]. Some websites use their own jargon, which refers to a daily language with abbreviations, acronyms and Anglicisms. This includes words such as LOL, TIC or emoticons that can lead to comprehension problems due to the difficulty that Down syndrome presents with abstraction.

#### 3.2.8. Textual Content Test 

A test was designed to determine the importance of understanding complex words to people with memory difficulties. According to the surveys carried out during the research, it is concluded that 23.2% of people with Down syndrome have comprehension problems always or almost always considering the slang commonly used during the navigation through the network. For example, when faced with words like Configuration, or abbreviations. Three tests are carried out for this:Unusual word + definition. Unusual word. Tests 16–17: an unusual word is included in a specific text of the two websites. A definition is offered on a website at the same time as it is needed (Figure 13). The definition is not included in the other website. It is concluded that it offers the definition of said word at the e moment in which it is needed, provides the user with a temporal gain of 68.4%.Abbreviated words. Test 18: the same procedure as in the previous test is followed, but abbreviations are used instead of complex words. In one web the abbreviation is included and in the other web the word without abbreviating is used. It is concluded that using complete words, instead of abbreviations, provides the user with a temporal gain of 75.5%.Memory test, explanation 30 min before. Test 19: people with Down syndrome are affected in the frontal lobe, responsible for linguistic development and memory. After analyzing the benefits of the previous vocabulary tests, a third test was carried out related to the reading level.

To know that the test is finished an action is proposed. The unusual words used in this test are an essential part of an activity statement. For example, “Configuration”. If the user does not understand their meaning, they will not be able to finish the test, so the objective is to seek a solution considering complex language so that certain users can navigate without needing to understand it. As solutions to this problem we propose the immediate explanation to the complex word, text links above the concrete word or use of a simpler vocabulary. This profile has anomalous formations in the speech device. These alterations they cause pronunciation problems of lingual consonants, lip consonants and rounded vowels. The morphology and neurological development of the phenotype with Down syndrome may be related to atypical patterns that negatively affect the development of specific language. And to the storage and phonological recovery skills. They get used to simple words because of their memory and pronunciation [26].

None (0%) of those tested knew the word Configuration. In Test 1 Website A 100% of users completed the test with the definition of the unknown word. In Test 1 Website B 52% of the users did not finish the test because no definition was added. A 75% of users finished the test in an average time of 128 s and a 45% of users took more time to finish Test 1 on Website B compared to the Test 1 on Website A. There is an average temporal gain of 277 s when a textual explanation is added at the same time as it is required, as opposed to explaining the word >30 min before. People with Down syndrome have memory and attention problems, which affect the retention of concepts in the short term [20]. For this reason, it is better that the definitions be included at the time they are needed and not before. The reduction of time in the test fulfilling the criterion 3.1.3 Unusual Words is 68.4% (Figure 14).

There were 76% of users who do not complete the test during navigation on Website B with abbreviations. There is a temporal gain of 394 s when words are not offered with abbreviations. Of the people who successfully completed the test that did not contain an abbreviation 60% had negative results in the test that offered an abbreviation. Abbreviated words are difficult for people with Down syndrome because of their abstract reasoning. The reduction of the time in the test fulfilling criterion 3.1.4 is 75.5% (Figure 15).

If the results are compared, some conclusions can be drawn. It must be said that this comparison includes the results of another test that will be explained later because we considered that its results are useful for textual content too. There was a 29% of users that needed complete words and not abbreviations. There was a 26.3% that needed immediate definitions of certain words. There was a 24.7% of images with low level of iconicity that are preferred to text. No users preferred a lot of text, (Figure 16):Using a Secondary Education reading level there is a temporal gain of 51.4%.By incorporating simple illustrations and with a low iconographic degree user can gain 64.3% of time.Incorporating too much text can affect comprehension. During the reproduction of a message aloud, there was a 94.1% of volunteers who ignored words that consider additional information. For example, in the phrase “click on the image when you see an orange”, the user remembers the word “orange” and forgets the others. He then responds to the oranges as soon as he sees them because he understands that the message is related to them. This does not mean however that the user understood the whole message.Users show a lot of frustration at incomprehensible words. There was an 86% that showed their frustration by gesticulating or emitting sounds of displeasure. This response can cause them not to finish their task, and of course, it causes their navigation to be unsatisfactory.

#### 3.2.9. Content Design Evaluation 

That the user knows what he is doing during navigation is important [8]. The cerebellum of a person with Down syndrome suffers from hypoplasia and a reduced number of granulosa cells, which affects visuospatial cognition, action activities and working memory. Working memory focuses on attention and operates in seconds, and for that reason, it is so important during navigation [23]. The alterations suffered by the parieto-occipital-temporal lobe affect visual recognition. This can cause difficulties in understanding many web design elements [25]. It is concluded, therefore, that the design of a website can be a decisive factor in its understanding. In addition, the reduction of the frontal lobe can hinder abstract or figurative thinking, which makes it difficult to understand an icon if it does not incorporate an explanation [16].

### 3.3. Content Design Test

This test is intended to determine the negative impact caused by a textured background. This test is carried out with eye tracking technology to analyze how the brain activity of volunteers responds to different web designs. We also wanted to determine if the labels are useful for people with neurological difficulties, such as Down syndrome, and if the non-textual content is an impediment or an aid to this profile.
Labels. Without labels. Tests 18–19: The aim is to quantify the positive impact of a label. To do this, iconography is used. Two tests are done, one without a label and the other with a label. In the main menu the options are proposed only with icons or icon + label (Figure 17). The time used to perform the icon comprehension test with and without a label is measured (Figure 18). This test is also done in a text, where abstraction is offered alone or explained. The time used to perform the text comprehension test with and without a label is measured.

The frontal lobe reduction of people with Down syndrome hinders their abstract thinking [24]. That is why it is difficult for them to understand icons if they are not explained. For example, non-figurative representations, arbitrary schemes, motivated schemes, pictograms or non-realistic figurative representations. In Website B there was a 29.4% of users who did not finish the test due to incomprehension of the icon. There was a 17.6% that does understand the same icon on Website B with a label. There is an average temporal gain of 1 min and 6 s by incorporating labels. The reduction of the time in the test fulfilling the guideline 1.1.1 Non-text Content and 2.4.6 Headings and Labels is 40.0% (Figure 19).
No need to use scrollbars for text or images. Scrollbars for text or images. Tests 20–23. The aim was to quantify the negative impact of using too many components on the webpage. There were two webs, one that presents a long content on a single page that forces the user to use the scroll bar (Figure 20). The other website presents the same content but spread over several pages. In this case, text links are used. It is verified that the user completes the test in two ways: (1) Observing whether the user uses the scroll bar or supposes that the content ends to up where it can be seen; (2) Verifying that the user understands the content. The content tells users that the test has ended, thanks there for their participation, explains the importance of their task, and asks them to leave to a specific room.

A 70% of users had difficulties using the mouse and 85% of users use text links. There is an average temporal gain of approximately 22 s during navigation if the scroll bar is bypassed. That is, if you avoid long text or a list of images. In both tests the gain is similar. This is because people with Down syndrome often have hypoplasia and orientation difficulties. That’s why they take more time. Breadcrumbs are a useful element for people with Down syndrome with some experience during navigation. However, it can become a distraction or an impractical textual component (Figure 21).

The reaction time in this trial may also be due to physical barriers, in addition to the neurological ones. From the results of the surveys it is concluded that 51.8% of people with Down syndrome present difficulties with the use of the mouse. Therefore, any measure that avoids large movements will be beneficial. The reduction of the time in the test fulfilling the guideline 2.4.4 Link Purpose (In Context), 2.4.5 Multiple Ways, 2.4.8 Location and 2.4.9 Link Purpose (Link Only) is 64.7%.Context does not change. Context changes. Tests 24–25. Context changes can mislead the user. The volunteer is asked to access a different page to continue with an activity. In one of the web pages this page respects the format of the previous one, the user knows that he is in the right place (Figure 22). On the other website the background, the typography and the structure of the new page are different from the ones in the statement and generate confusion, (Figure 23).Five element menu. Seven element menu. Tests 26–27: Understanding each element of a menu is difficult for a person with difficulties in abstraction. This test has two menus: in one there are five elements and in the other seven. It is intended to find out if the number of elements affects understanding. The test is shorter than the previous ones. The user is asked to locate an element and must point to the element.Monochromatic backgrounds. Textured background. Tests 28–29: The aim is to quantify the negative impact caused by a textured background on a person with Down syndrome. To do this, users are asked to locate a certain phrase on two websites, one with a white background and the other with a textured background, see Figure 24. The times of the four texts indicated in the images are timed as T1, T2, T3 and T4.

There is an average motivational indicator of 33 s stronger against a monochromatic background color because it does not interfere with the legibility of the text. Less eye movement is observed in the path of Website A (monochromatic background) than in Website B (textured background). During the tracking of Website B users registered greater brain activity. Probably when viewing so many elements of distraction the user need more concentration. The reduction of time of the test fulfilling the guideline 1.4.1 Use of Color, 1.4.6 Contrast (Enhanced), 1.4.8 Visual Presentation and 3.2.2 On Input, is 65.0% (Figure 25).

By comparing all content design trials, some conclusions can be drawn. A 26.9% of users use and understand the color signals. There is a 24.50% of users who surf better with a monochromatic background. There is a 22.6% who use labels. There is a 16.7% who prefer menus with less than 5 items. There is a 6.0% that does not present problems with a change of context. There is a 3.3% that has no problems with scroll bars (Figure 26):
There is a stronger motivation indicator in a web with a monochromatic background. The temporal gain when the bottom of the web is monochromatic is 65.0%.The color signals that highlight or transmit information are beneficial for the user with neurological deficit. The volunteers gained 71.4% of time when the content offered informational colors (not meaningless colors).Eliminating unnecessary elements in a webpage keeping only the menu, the toolbar and the content (with title and navigation routes), helps to understand the web. A 32% of volunteers did not finish the tests in the web pages that contained more than five elements.When there are design changes or content changes there is a 41% chance that the user will be wrong.There is a 68% of volunteers who show difficulties with the mouse, especially when there are scroll bars.The incorporation of textual content or informative labels reduces browsing time by 40.0%.

### 3.4. Form Evaluation

Completing a form effectively requires numerous neurological activities, such as language and communication (during the reading of the statement and response), short-term memory (during the response), visuospatial ability (during the understanding of the structure) [25,27], and association (during the reasoning of the options) [25,28]. The areas of the brain responsible for these functions are the frontal lobe, the temporal lobe, the cerebellum and the parieto-occipital-temporal lobe. All of them are affected in people with Down syndrome [25].

#### Form Design Test

There were five trials on one website for this test. A user who has a deficit in working memory, in abstraction and in comprehension, is very likely to have great difficulty in filling out a form. This test is designed to determine which are the most understandable form fields for users with Down syndrome. Simple questions and different response methods are offered. The time spent, the completion of the test and the response offered are valued to know which the optimum form field is. The aim is to verify what the needs of a user with Down syndrome are when he answers forms. The aim is to verify what the needs of a user with Down syndrome are when he completes forms:Click field 1 cm. Click field 0.5 cm. Tests 30–31: the objective is for the volunteer to answer a question by choosing a response from among a group of responses. To choose the answer, the volunteer must click on its circumference (Figure 27).Help text disappears. Test 32: the objective is for the volunteer to respond freely in an empty field. An out-of-field instruction is offered and, in addition, a help text in the same field, but this help text disappears when the volunteer clicks inside it.Drop-down field. Test 33: the objective is for the volunteer to respond through a drop-down menu.Empty field. Test 34: the objective is for the volunteer to respond freely in an empty field. The question offers the instruction in the same field, for example “Name”, but that instruction disappears when the volunteer clicks inside it.

By comparing all the form field design trials, some conclusions can be drawn. Only 19.5% of users successfully hit in click fields if the space is less than 1 cm. Only 16.5% of users answer the questions if they do not see the statement or if the statement disappears. Only 15.0% of users could not hit in click fields if the space was less than 0.5 cm. Only 13.6% of users use the drop-down field quickly. There is a 14.8% that demands that there be a preview before a field. For example, in a configuration form. There is a 12.4% of users who read all the options offered to possible answers. There is 8.2% of users who manage to answer the questions even though their statement disappears when they click on them (Figure 28).
When a user must click on a circle, he needs precision. The larger the circle, the less the risk of failure. A 75.9% of the volunteers correctly clicked the 0.5 cm circle, and 98.5% of the volunteers correctly clicked the 1 cm circle.Fields that include visible instructions and help text offer a temporal gain of 83.3%.The forms with drop-down menu have a high level of abstraction and that makes it difficult to understand. There is a 30.5% of volunteers who cannot click on the answer when the menu is drop-down.An empty form field that offers instructions inside is an advantage for the user. When these instructions disappear when you click inside, the temporal gain is 41.7%.A form with a list of visible answers offers a temporal gain of 62.5%.Not including too many options helps users with Down syndrome to choose an answer more easily.The configuration of the form is useful for the user except when the user does not understand the changes that may occur. A preview helps to understand possible changes and reduces browsing time by 74.8%.

### 3.5. Color Contrast Evaluation

The parieto-occipital-temporal lobe of Down syndrome is responsible for visual recognition and undergoes alterations in the microstructure of the cells. This problem causes both colors and shapes to be confusing [25]. This does not mean that the visual problems caused by the profile are serious, but visual interpretation can be an inconvenience [6,25]. In addition, Down syndrome is characterized by attention problems. The incomplete development of the cerebellum affects working memory, which focuses on attention [20]. A visual misinterpretation can cause failures and result in sub-par website browsing for people with Down syndrome [20,25].

#### Color Contrast Test

There were four trials on one website for this test. This test offers some websites with several types of background color, button and text. We want to know which the most optimal contrast is and if there are colors that especially distract the attention of volunteers. The test asks the user to find four specific elements within the page. The user must look for the same elements on all the websites. In this way avoiding alteration in the objectivity of the results, because it may be that the user tends to look first at one side of the page. Or it may be that the user always traces with the same pattern. The time spent in finding the button demanded is measured. In addition, this test is performed with rest intervals and interspersed in other tests, to prevent the user from remembering the answers.

Five trials are carried out to test different background colors and button colors (Figure 29, Figure 30, Figure 31, Figure 32 and Figure 33). The aim is to find out which contrasts are more suitable for users with Down syndrome. Eye tracking technology is used for these tests.

One can consult the statistics of preferences according to the contrast between background and letters in Figure 34 and Figure 35:A blue foreground color # 3B29FA on a white background #FFFFFF (contrast 7.28:1) causes an attention span of 1891 s.A close-up white color #FFFFFF on a blue background # 3B29FA (contrast 7.28:1) causes an attention span of 2925 s.A color of foreground cyan # 00FFFF on a black background # 000000 (contrast 16.75:1) causes an attention span of 4270 s.A close-up cyan color # 00FFFF on white background #FFFFFF (contrast 1.25:1) causes an attention span of 6533 s.A black foreground color # 000000 on a yellow background # FFFF00 (contrast 19.56:1) causes an attention span of 7246 s.

### 3.6. Links Evaluation

The Internet demands some autonomy to reduce reaction times [25,29]. As with the color tests, attention deficit and problems during visual interpretation may affect the understanding of text links [16]. A text link can be interpreted as meaningless by a user with Down syndrome. This would be a distraction.

#### Link Test

There were four trials on two websites for this test. On the homepage of both websites several tests are listed. The user must to go to one of the tests. The test that is requested is chosen arbitrarily. In one of the websites the user is forced to search for the demanded test in the menu because the statement does not contain text links. On the other website there are text links. The text link is found in the statement of the test, so that the user can click on the text link at the same time they are asked to go to the test. The test is concluded when the user arrives at the intended page (Figure 35). The objective is to determine if the links are perceptible and useful for users with Down syndrome, with neurological and motor difficulties (Figure 36). The cerebellum of Down syndrome subjects has hyperplasia, and this causes one to have orientation difficulties that prolong your response time when clicking.

It can be concluded that there is a 35.0% of users who do not navigate without text links to the rest of the route options. There is a 26.1% that does not understand its location without breadcrumbs. There is a 15.1% that cannot crawl with the mouse (Figure 37).
According to the surveys, there is a 51.8% of users with Down syndrome with difficulties using the mouse.It is beneficial for navigation to avoid large movements. Links within the context provide a temporal gain of 64.7%.That a website shows information about the location offers a temporal gain of 71.8%.

### 3.7. Temporal Elements Evaluation

As with color, visible elements that limit time increase distraction in people with Down syndrome. A chronometer distracts users with Down syndrome because its movement distracts their attention. This is so because the cerebellum causes the action activities to focus on the most prominent visual elements, which may be those that highlight the temporal elements. In addition, the deficit in working memory makes it difficult for navigation actions to end. The deficit of working memory is due to the reduction of the frontal and temporal lobe [25].

#### Temporal Elements Test 

There were two trials on two websites for this test. In both there is a simple questionnaire where users must choose several options simply by clicking. In a web this questionnaire is accompanied by a countdown (Figure 38 and Figure 39). On the other website there is no countdown. The objective is to quantify the negative impact of visible temporal elements. Stopwatches, countdowns and interruptions are considered temporary elements. For the test, two websites are proposed, one with a visible chronometer and the other with an invisible chronometer. The size of the visible timer occupies 15% of the form to ensure that users see it. The stopwatch has a pause button.

There was a 13.3% of users who were able to finish the test with a visible chronometer. However, during the test without time limits all users finished. There is an average temporal gain of 104 s when the form does not limit the browsing time. The reduction of time of the test fulfilling the criterion 2.2.1 Timing Adjustable, 2.2.3 No Timing, 2.2.4 Interruptions, 2.2.5 Re-authenticating is 52.5% (Figure 40).

By comparing all the results, several conclusions can be drawn. There is a 40% of users who cannot finish a task if there is a stopwatch that does not allow pausing. There is a 37.6% of users who do not finish the task if they cannot prolong the time. There is a 21.9% of users who do not finish the test if there is a temporary element visible (Figure 40).
Without visible temporary elements, volunteers gain 52.5% of the time.There is 97% of the volunteers who press the PAUSE button, stating they need to stop the stopwatch.In trials where you cannot stop the time, 90% of users did not finish.

## 4. Conclusions and Recommendations

### 4.1. Recommendations

From the results of the user tests, a series of useful recommendations for web developers is presented. These recommendations are divided into:-Recommendations for playing multimedia elements-Recommendations for audio files-Recommendations for presenting the content-Recommendations for the design of forms-Recommendations based on color contrasts-Recommendations for links

These recommendations are based on user tests performed with people with Down syndrome. However, it is important to note that these recommendations are also applicable to other user profiles. People with Down syndrome have a set of symptoms that are also present in other users. For example, the explanation of unusual words in the text itself is useful for people with Down syndrome and for any other user with learning problems, or for users who do not know the language, or even people with low vision who use a screen magnifier [30]. In [30] W3C includes a list of possible beneficiaries for each of the success criteria.

### 4.2. Multimedia Elements Recommendations

(1)During the tests of reproduction of multimedia files, it is concluded that it is preferable to use visual elements instead of auditory ones. This is due to the auditory attention deficit of the subject profile. It is recommended to avoid sign language or iconic language. This recommendation may contradict accessibility guidelines 1.2.3, 1.2.5 and 1.2.7 because the reproduction of auditory content may be abstract for users with Down syndrome. In the subtitle tests, we observed that subtitles and audio are more efficient at the same time. When there is more than one form of communication, it can confuse the user. This recommendation may contradict accessibility guideline 1.2.2 and 1.2.4 because the reproduction of audio and text at the same time may confuse a user with attention deficit and with short-term memory alterations. However, when an audio file does not include a visible narrator, it may be useful to include subtitles.(2)It is not recommended to play very long audios. It is also not advisable to add additional visual information to the audios, when said information is complementary and does not describe the main message. This recommendation may contradict criterion 1.2.7 because it is not advisable to offer two types of messages at the same time to people with short-term memory deficits(3)We observe that it is not recommendable that the multimedia files are reproduced automatically without the consent of the user. When this happens, users lose concentration and his deficit in short-term memory does not allow them to retain the content of the audio that was reproduced without warning(4)It is recommended to incorporate pause and restart buttons. During the navigation tests on the accessible website, users used these buttons. During the inaccessible website browsing, users clicked on the screen waiting for the audio to stop.

### 4.3. Sounds Recommendations

(1)It is recommended to play audio files with a volume between 20 dB and 50 dB. A person with auditory deficit hardly listens to volumes below 20 dB and needs a lot of concentration to understand the message. A volume greater than 50 dB is a considerable auditive distraction and is not effective for the transmission of a message.(2)It is recommended that background sounds be four times quieter than the main one. It is observed that if this is not fulfilled, there is a greater possibility that the message goes unnoticed.(3)It is recommended to add “pause” and “restart” buttons. During the navigation tests on the accessible website, users used these buttons. During the inaccessible website browsing, users clicked on the screen waiting for the audio to stop.(4)It is recommended that the content of the messages be understandable and forceful.

### 4.4. Textual Content Recommendations

(1)The level of language used is important for both comprehension and motivation. In the broadcasts where we use complex words, abbreviations or technicalities, the volunteers dislike the broadcast. It is recommended to avoid unusual, complex or technical words.(2)In the case of using unusual words, it is recommended to provide a textual definition within the same web page. This definition can be linked to a text link, within the same text.(3)It is further recommended that the definition be displayed while the meaning of the unusual word is required (Figure 41).

(1)It is recommended to avoid abbreviations as much as possible.(2)If abbreviations are used, it is recommended to provide a textual definition through a link incorporated within the same web page (Figure 42).

(1)For the same reason it is recommended to use a Secondary Education reading level, with short and consistent sentences of less than 25 words. It is recommended to avoid jargon, abstract terms and redundancies.(2)It is recommended to avoid literary resources that can be confusing for the user. To generate this recommendation, we used metaphors in one of the tests. The volunteers did not understand the metaphor, so it is concluded that a literal language is preferable.(3)In the iconicity test, the volunteers were interested in the illustrations, however, they did not show much understanding of them. Their results varied depending on the degree of iconicity, being the icons with low level of iconicity and the recurring icons or with descriptive labels simpler. It is recommended to include illustrations with low level of iconicity. This recommendation may contradict the usability criterion that indicates that the use of images increases speed during navigation. It can also contradict the usability criterion that indicates that it is favorable to use standardized icons that replace the text, such as error, success and warning notifications.(4)In all the tests two fonts were offered: one textured and one monochromatic. All the results obtained were considered. It is recommended to use black text on a white background.(5)Several tests on subtitles were carried out. It is recommended to use a background color that highlights the text. In dialogues between several people it is recommended to use different colors with sufficient contrast.

### 4.5. Form Design Recommendations

(1)The proof of forms was very enlightening, since the effectiveness of the design of the field of a form serves in the same way from a menu or for other functions. The most effective response fields are, in order of preference: Elements to clickEmpty fields with visible instructionsExpansion menusEmpty fields with visible instructions until clicked

This recommendation may contradict the usability criterion which indicates that it is preferable to limit the initial explanations in the forms. It also contradicts the recommendation that if the explanation is necessary, it could be added using a link. For users with Down syndrome it is recommended that the fields of the form include initial explanations. In addition, it contradicts the usability criterion that indicates that the information requested must be limited, for example, by hiding optional parts or using expandable forms. For a user with Down syndrome with motor and neurological limitations, it is recommended not to use expandable menus or hidden options that may go unnoticed.
(2)It is recommended that the configuration forms include a list of predisposed options. For example, a menu to change the background color with various color combinations. That series of options cannot have more than five options.(3)It is recommended to offer previsualized configuration forms. It is recommended to use two previsualized, one previewed with the same button (for example a button with red background and white letters to change the colors of the web to red and white) and a second one previewed in the webpage itself so that the user understands what the change consists of.(4)It is recommended to try the changes at the request of the user do not imply a change of context.(5)It is recommended that the options of the forms have a size proportional to the visible space of the web page to avoid the use of scroll bars.

### 4.6. Contrast Recommendations

(1)The color contrast test was performed using eye tracking technology. For this, it was necessary to offer different pages of the test website and wait for the retina of each volunteer to indicate the most striking aspects. This test is interesting, because Down syndrome is considered a profile with an acute attention deficit. The results of this test show those contrasts that kept your attention longer. The most effective color contrasts are, in order of effectiveness: blue button # 3B29FA on a white background #FFFFFF, white button #FFFFFF on blue background # 3B29FA, cyan button # 00FFFF on black background # 000000, cyan button # 00FFFF on white background # FFFFFF, black button # 000000 on yellow background # FFFF00.

### 4.7. Link Recommendations

(1)For the test of links, different texts were offered with tests that invited users to access different pages related to the content of the text. It is recommended to incorporate the links within the context and include their purpose (Figure 43).

This recommendation may contradict the usability criterion that states that if the content is too long, text links should be included. For example, with a link that says, “keep reading”. This type of additional information may be abstract for a user with Down syndrome. It is recommended that the texts do not contain extended information because attention deficit and short-term memory make comprehension difficult. Text links are useful when their purpose is to help the user in their location, for example by clicking the “Cart” button. It is also recommended to avoid having an additional information redirect to a different page.
(2)It is recommended to add links to navigate to related web pages. This recommendation guarantees that users know how to get to where the page invites them to arrive. During one of the tests, additional information was offered to complex texts. In the accessible web, they were offered a link that led to a page that explained the definition of the complex world, or the explanation of the purpose of the page (according to the test). It was shown that these additional links are useful as long as their usefulness is explained.(3)It is recommended to use breadcrumbs to offer information about the user’s location. This test was like the previous one, but the text statement invited the user to change the page to a specific page (Figure 44).

### 4.8. Temporal Elements Recommendations

(1)In this test, a visible temporal element was placed in one of the test websites. A high percentage of volunteers could not finish the test due to the level of distraction offered. It is recommended to avoid unnecessary temporary elements.(2)It is recommended to offer the option to pause or prolong the time. The volunteers try to stop the chronometer by pressing the page, so it is concluded that a button that allows the pause or prolongation of time would be effective.(3)It is recommended to offer the option to deactivate or postpone interruptions, except those that imply an emergency.(4)It is recommended that the user loses data when his session expires.

### 4.9. Labels Recommendations

(1)The descriptions were useful in all the tests, because the volunteers on multiple occasions need help to understand the tasks or the functions of a web. It is recommended to incorporate labels to the icons.(2)It is recommended to add descriptive headers.(3)It is recommended to add descriptive labels (Figure 45).

## 5. Conclusions

For website accessibility to be effective, it is recommended to analyze the user profile to which the web is directed. For this, their physical and neurological characteristics must be studied, relating each of them with the recommendations on accessibility and usability to verify their effectiveness. There are many studies that analyze through surveys the existing web needs of users with Down syndrome and users with cognitive disabilities. These surveys are usually directed to a very wide age range. This type of study is very useful to raise awareness in society and about the importance of inserting these people in activities that require the use of computers. However, this type of study may have limitations. The first limitation is the age range that is usually proposed. When they are studies directed to students as they begin to analyze the use needs of 4 years old. At this age many people with cognitive disabilities have not yet acquired reading and writing skills, necessary during navigation.

The second limitation is to try to meet these needs without having proposed a test of them. Users with cognitive disabilities are not always aware of their limitations or their needs. It is possible that some of the questions answered by people with Down syndrome offered a non-objective or unrealistic response. For example, the study carried out for this research shows that 42% of the volunteers do not present attention problems. This response is not objective considering the attention deficit that characterizes the profile. This is demonstrated in several of the tests carried out in which a considerable number of volunteers leave the test unfinished because there is a visual element that distracts their attention, but there are questions that can be very useful in terms of opinion. For example, we were interested to know if users with Down syndrome often modify the configuration of their computers, or if they use the help button, or if they feel isolated in activities that require a computer. In this research, a survey was conducted that sought to obtain these data, in addition to others related to the needs that users have consciously or believe they have. The survey had no age limit because one of the objectives was to know which age range is more interested in new technologies, which was deduced to be that between 15–35 years old.

On the other hand, there are also accessibility and web usability guidelines aimed at people with cognitive disabilities. However, there is currently no research that tests and evaluates these guidelines. This was the number one priority in the research, since it is considered necessary to know if the guidelines that currently exist are effective. During the tests we found some contradictions between accessibility and usability guidelines and the results of our tests. For example, we found a contradiction between criterion 1.2.3 “Description of pre-recorded audio”, 1.2.5 “Audio description” and 1.2.7 “Extended description” and the results of the tests on multimedia reproductions. Our results indicate that audio reproduction can be abstract and difficult to understand, so any auditory allusive information would be ineffective. It is concluded that including a visible narrator to an audio file is 86.1% faster than a single audio file. An audio file below 20 dB or above 50 dB causes a time loss of 82.7% compared to an audio reproduced in that interval.

Guideline 1.2.2 and 1.2.4 “Subtitles” also contradict each other, since our results conclude that the projection of a moving text can be confusing for people with attention deficit and short-term memory problems. On the other hand, it should be noted that a text message is more useful than an auditory one, as long as the text is short and direct. It is demonstrated that by eliminating audio and conserving subtitles, navigation is 2.1% faster than if audio and subtitles are played at the same time. In usability the guideline that states that the use of images can be useful for the speed of performance is contradicted, since we verify that the icons are difficult to understand. In the same way, the criterion that indicates that the information must be limited by means of expandable forms or by links to other pages is contradicted. In the research it has been verified that this type of fields is confused and not very perceptible for users with Down syndrome. Regarding text links, they must be incorporated into the text itself. It is determined that in a drop-down field requires greater concentration because you need to display the hidden menu and click on one of the answers. There is a 68.6% of the volunteers that displays the menu and does not click on any answer. It is also determined that adding labels to the icons reduces the reaction time by 40%.

In this research, contradictions have been observed between the WCAG 2.0 guideline and the usability guidelines and the results of the study tests. That is, not all existing guidelines are effective for users with Down syndrome. In addition, a series of useful accessibility and usability recommendations for people with Down syndrome are extracted. All tests, including surveys, were conducted with people with Down syndrome. The volunteers did not receive personal assistance during the tests or during the surveys. For this purpose, previous tests were carried out that determined the design, the wording and the level of difficulty most appropriate to the profile. The environment for the user tests was made based on the needs in the results obtained in the surveys. WCAG 2.1 has incorporated new guideline. Although the approval of the new guidelines is after our study, some approach our conclusions. For example, guideline 1.3.5 that suggests the possible benefits of a simpler webpage structure resembles our conclusions regarding the comparison of the two websites proposed for the study. Criterion 1.4.10 that aims to dispense with the scroll bar matches our results in terms of the content structure. The criterion 1.4.11, in contrast, that can be verified in our tests with eye tracking technology. Criterion 1.4.13 regarding hidden information can be linked to our tests with form fields.

In Figure 46 we can see the time in seconds spent on the tests. These results are an average of all the trials. Each trial contained several tests to determine if the results were useful. The result of each of the trials is found in the paper itself. The longer time spent was less effective. One can compare the time spent on the two websites, the accessible website A and the inaccessible website B. A considerable temporary gain is observed in all the tests that are intended to be accessible. Time is important during website browsing, so we can conclude that following these proposed recommendations will offer an advantage to users with Down syndrome. Disability is a term that evolves, that depends on the context and that does not imply perpetuity. It is a relative term, so it depends on the web developers that there is a digital context in which there is no lack of capacity.

This research has helped us understand the importance of knowing users well to meet their needs. All the tests have been designed specifically for this profile and the results have been favorable. Many of these tests are present in daily activities, such as filling out a form to request a medical consultation. For future work we would like to try the same tests but with daily activities. For example, analyzing websites of the administration, social networks or leisure. Knowing how our trials positively affect daily activities would be important for our research.

## Figures and Tables

**Figure 1 sensors-18-04047-f001:**
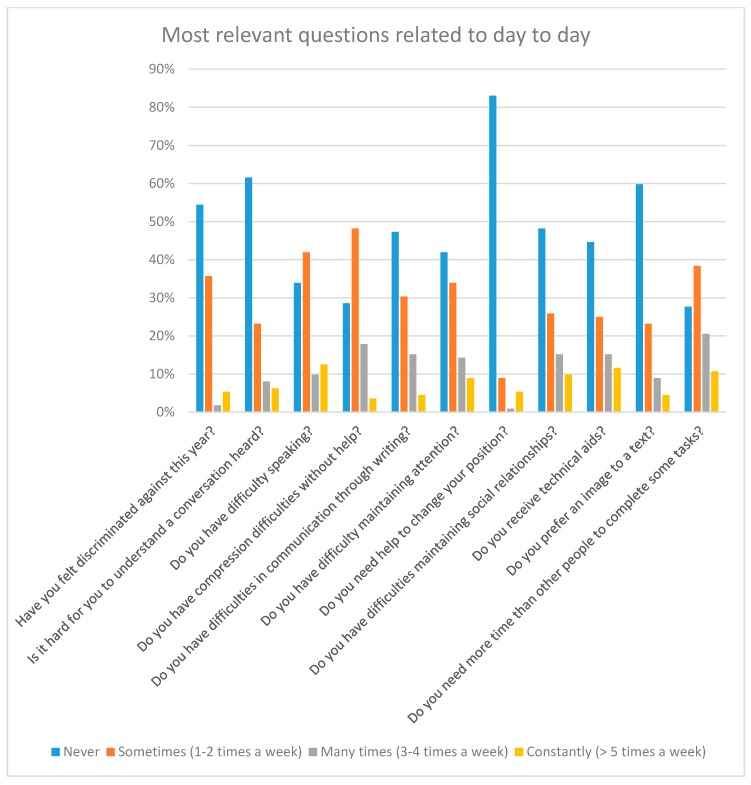
Most relevant questions related to daily activities.

**Figure 2 sensors-18-04047-f002:**
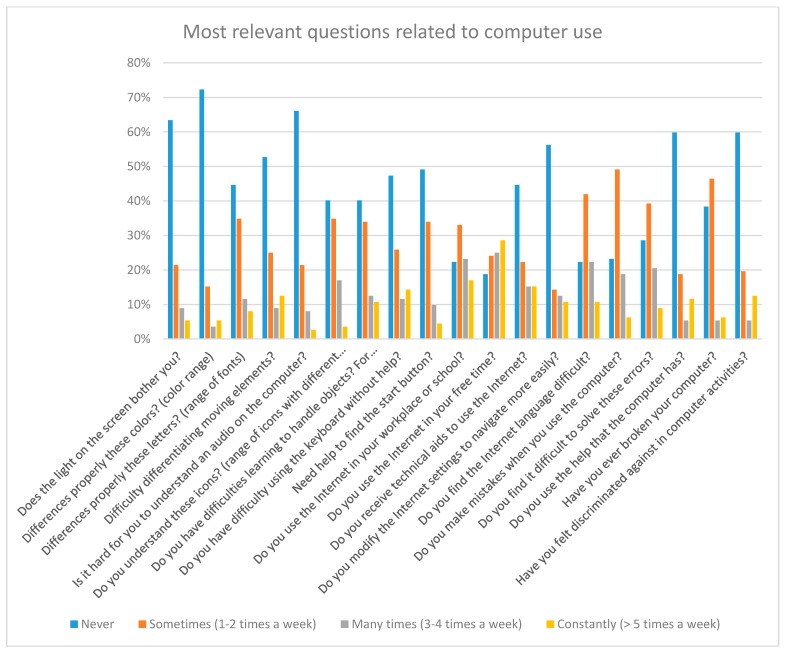
Most relevant questions related to computer use.

**Figure 3 sensors-18-04047-f003:**
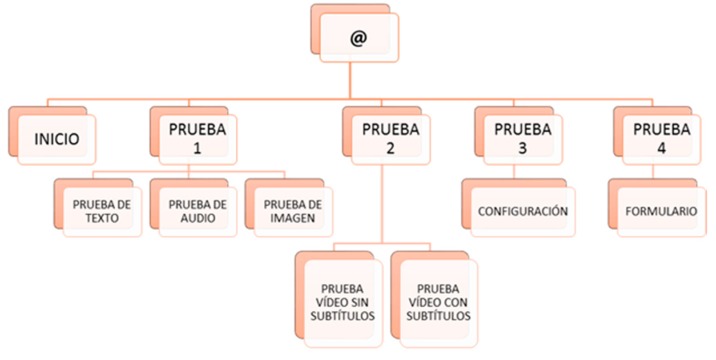
Structure of the web user test.

**Figure 4 sensors-18-04047-f004:**
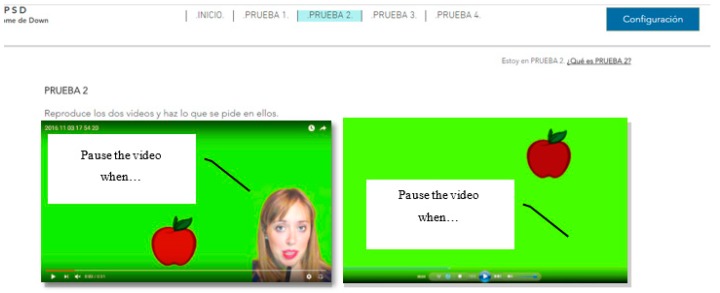
Multimedia test. Audio with and without narrator. Without subtitles. “Pause the video when you see an apple”.

**Figure 5 sensors-18-04047-f005:**
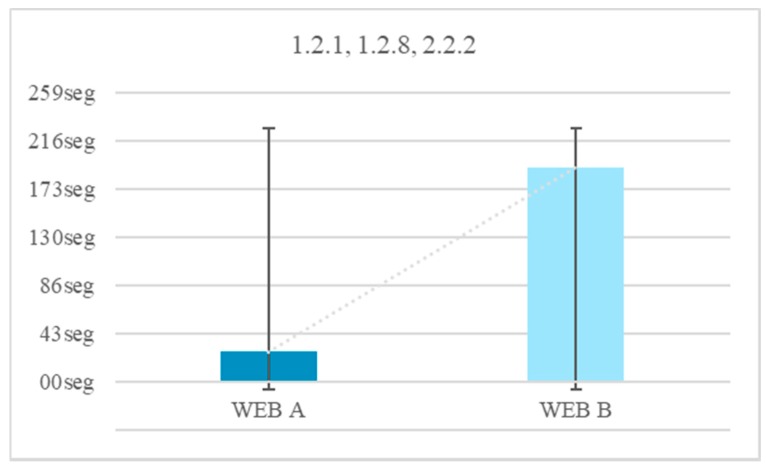
Audio + video + narrator. Audio + video. Test 1–2 results.

**Figure 6 sensors-18-04047-f006:**
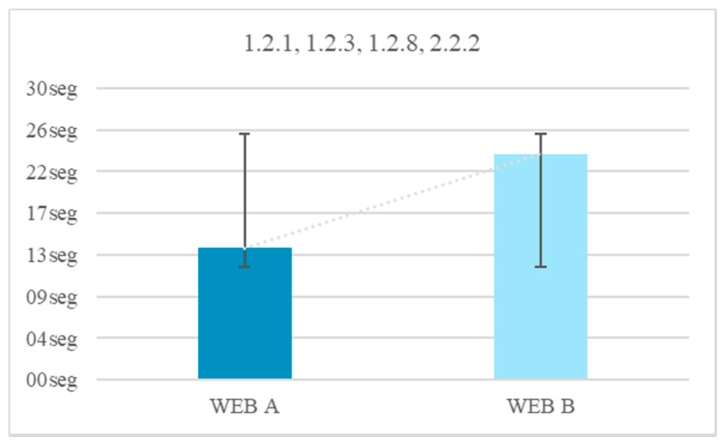
Audio + subtitles + video + narrator. Audio + subtitles + video. Tests 3–4.

**Figure 7 sensors-18-04047-f007:**
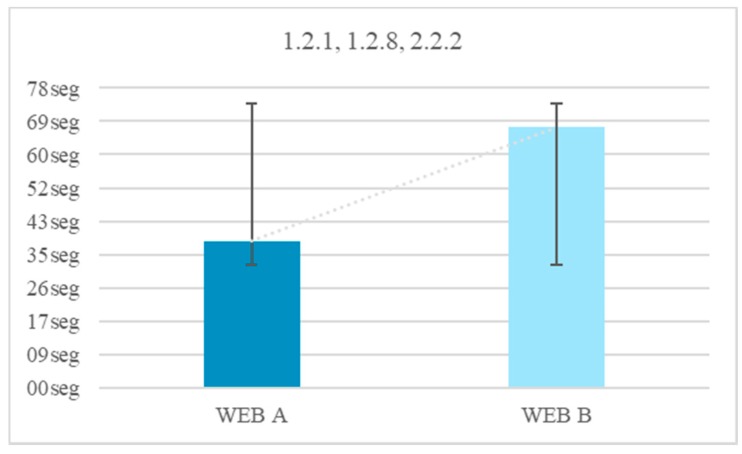
Video + subtitles + narrator. Video + subtitles. Tests 5–6.

**Figure 8 sensors-18-04047-f008:**
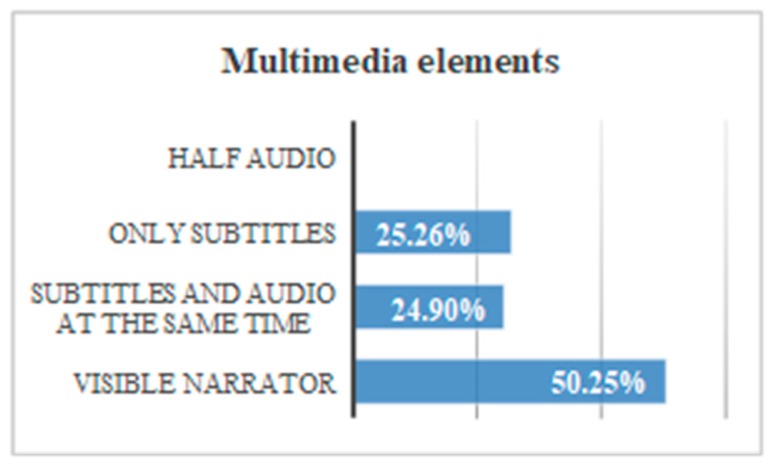
Evaluation 1.

**Figure 9 sensors-18-04047-f009:**
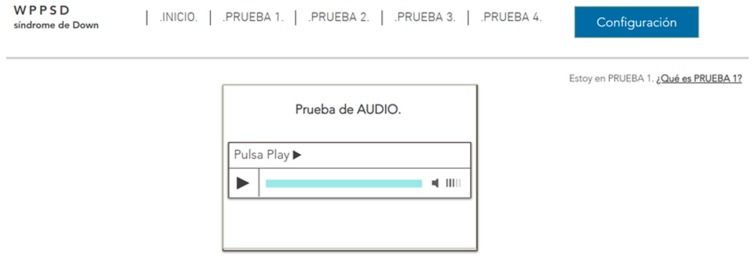
15 dB message with a slight background noise with sound adjustment tools.

**Figure 10 sensors-18-04047-f010:**
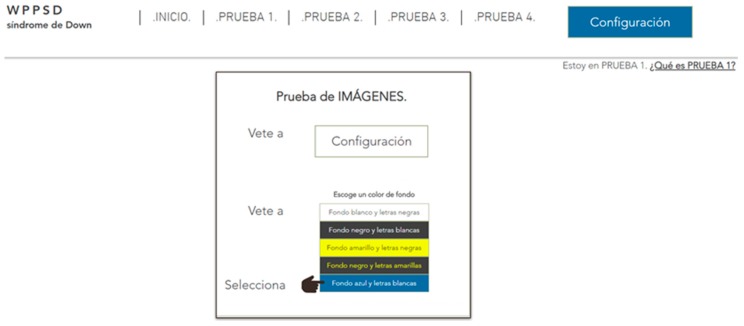
Audio message: “Go to configuration and select the color (…)”.

**Figure 11 sensors-18-04047-f011:**
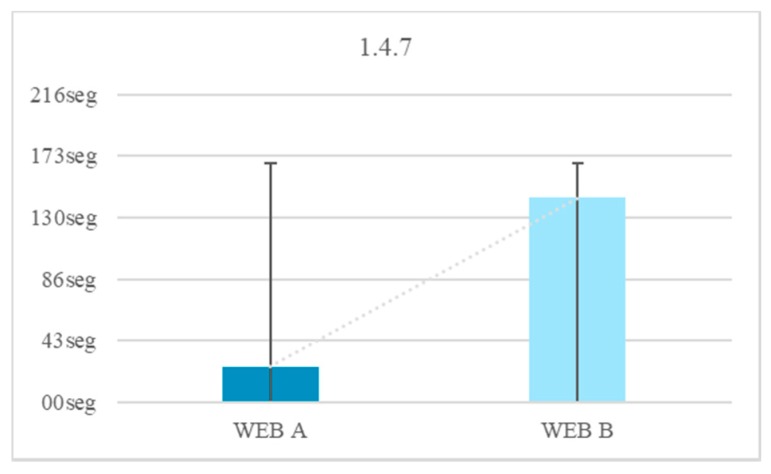
Tests 14–15. Website A: 20 dB without a slight background with sound adjustment tools. Website B: 20 dB without a slight background noise without sound adjustment tools.

**Figure 12 sensors-18-04047-f012:**
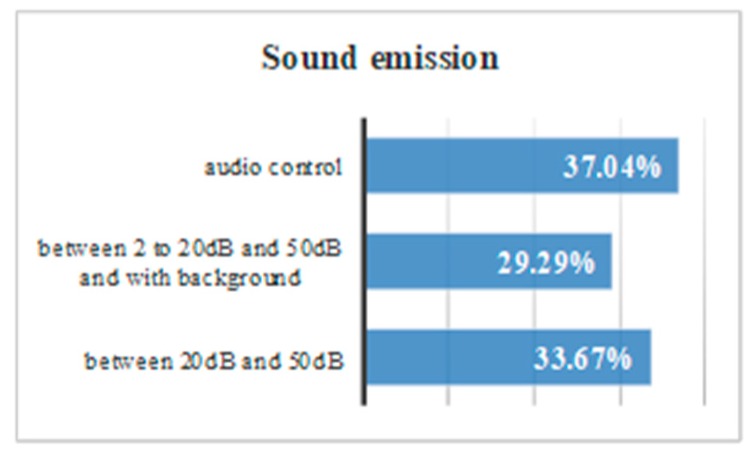
Evaluation 2.

**Figure 13 sensors-18-04047-f013:**
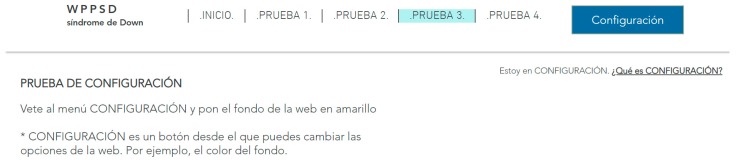
Unusual word + definition. Test 16.

**Figure 14 sensors-18-04047-f014:**
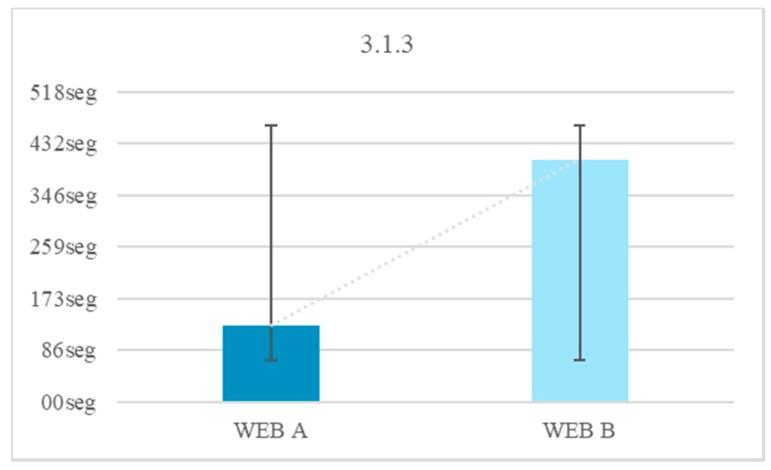
Unusual word + definition. Unusual word. Tests 16–17.

**Figure 15 sensors-18-04047-f015:**
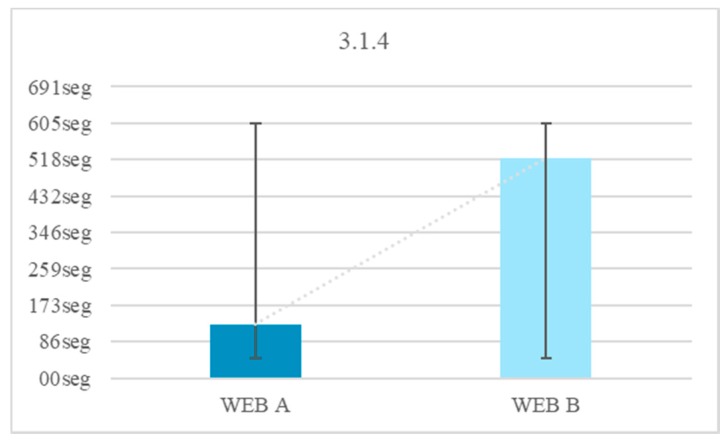
Unusual word + definition. Abbreviated word. Tests 16–18.

**Figure 16 sensors-18-04047-f016:**
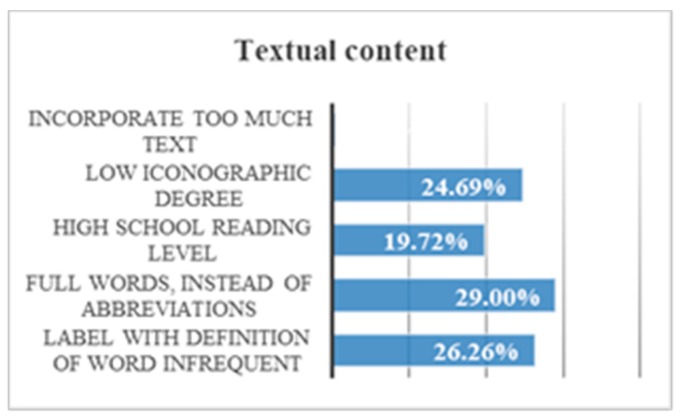
Evaluation 3.

**Figure 17 sensors-18-04047-f017:**
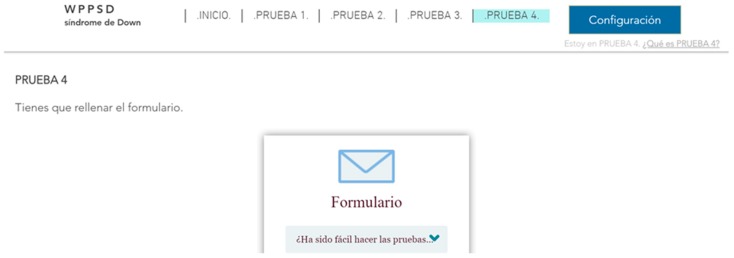
Label on website A icon. Test 18.

**Figure 18 sensors-18-04047-f018:**
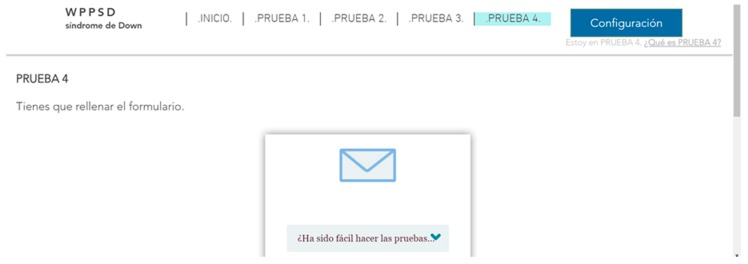
An icon on Website B without a label. Test 19.

**Figure 19 sensors-18-04047-f019:**
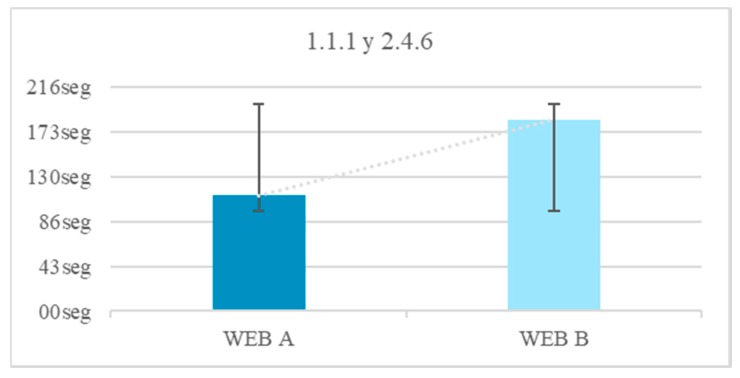
Labels. Without labels. Tests 18–19.

**Figure 20 sensors-18-04047-f020:**
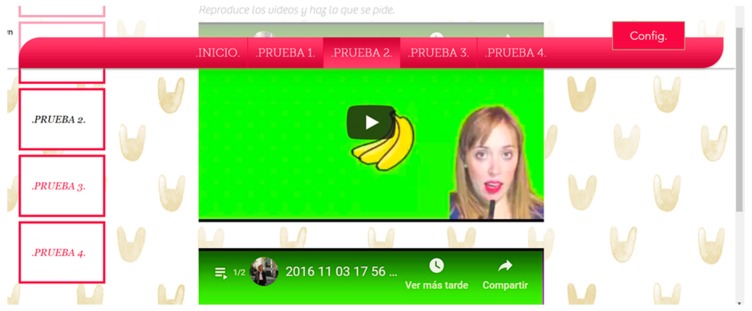
Website B needing scrollbars. Images instead of text. Test 23.

**Figure 21 sensors-18-04047-f021:**
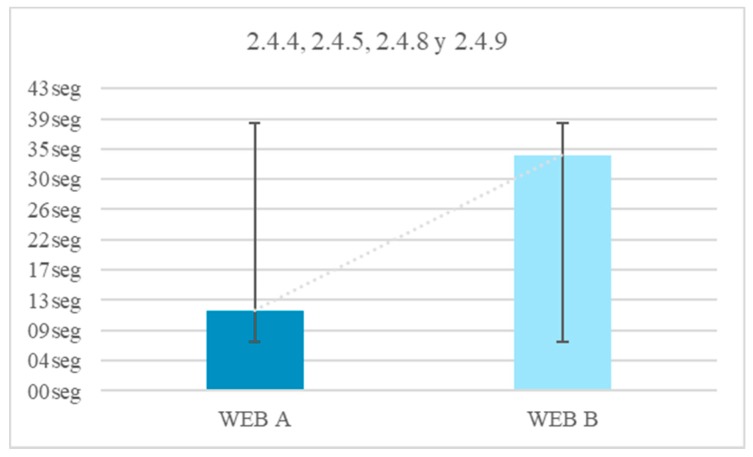
Without need to use scrollbars, images. Scrollbars, images. Tests 22–23.

**Figure 22 sensors-18-04047-f022:**
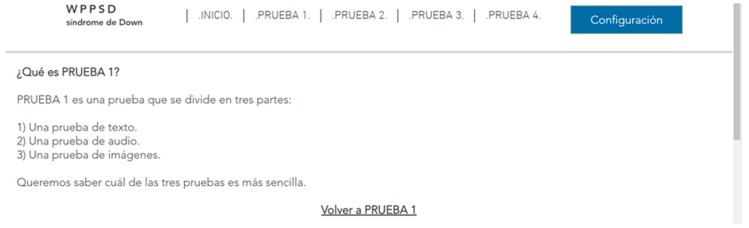
Statement context 1. Tests 24–25.

**Figure 23 sensors-18-04047-f023:**
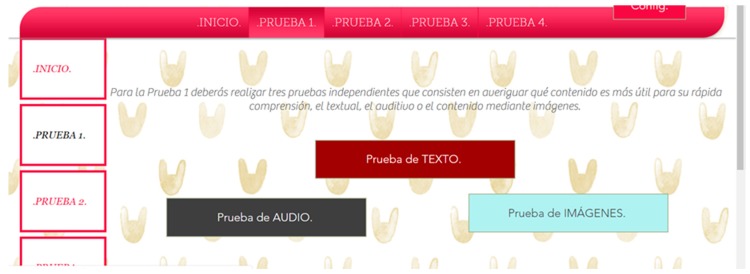
Context change. Test 25.

**Figure 24 sensors-18-04047-f024:**
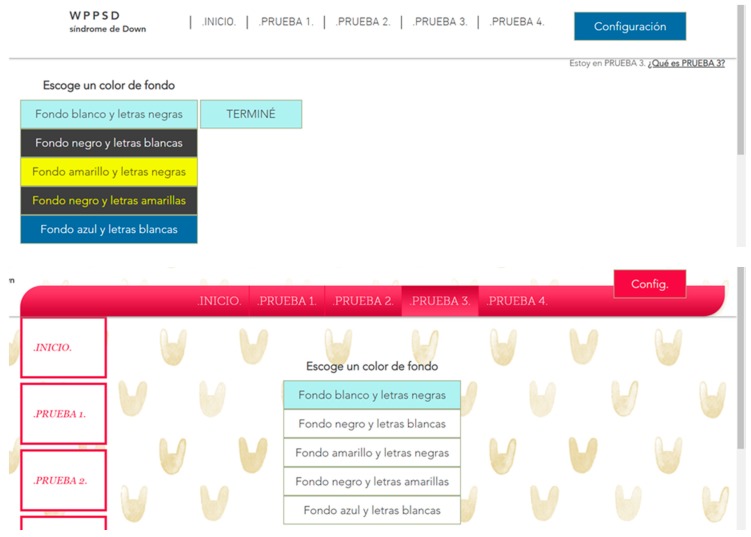
Test backgrounds on Website A and Website B.

**Figure 25 sensors-18-04047-f025:**
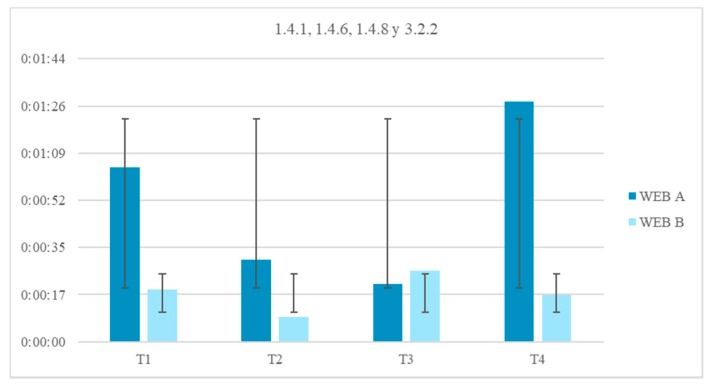
Monochromatic background. Textured background. Tests 28–29.

**Figure 26 sensors-18-04047-f026:**
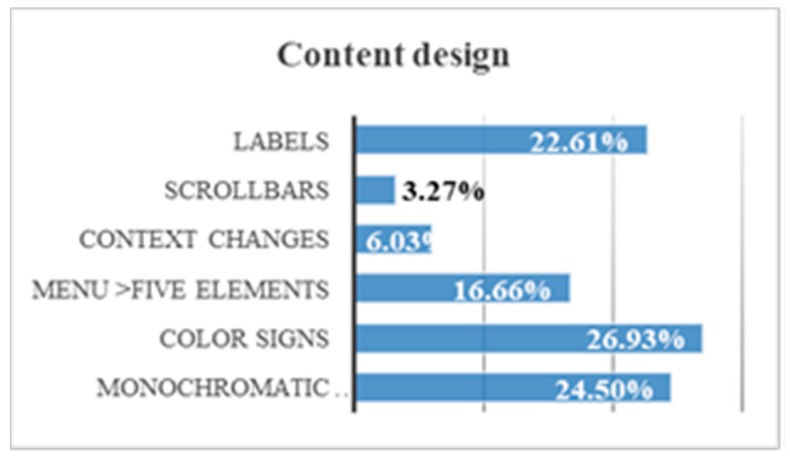
Evaluation 4.

**Figure 27 sensors-18-04047-f027:**
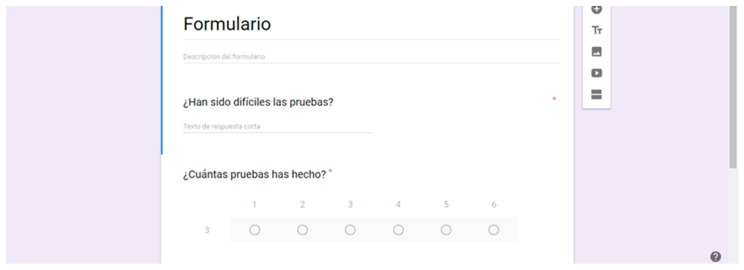
Click field 1 cm. Click field 0.5 cm. Tests 30–31.

**Figure 28 sensors-18-04047-f028:**
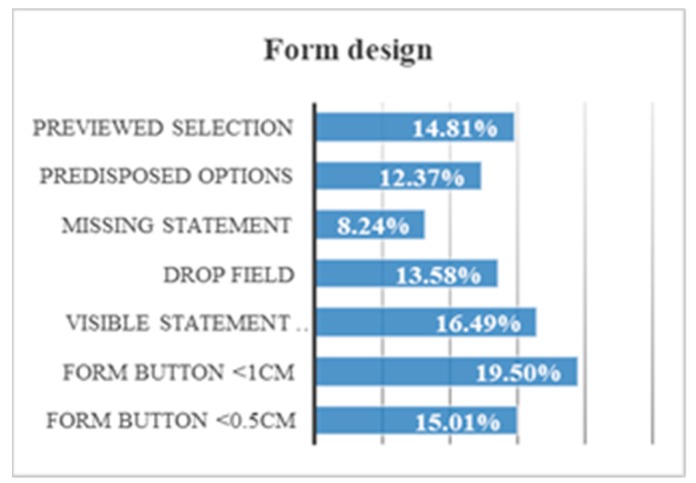
Evaluation 5.

**Figure 29 sensors-18-04047-f029:**
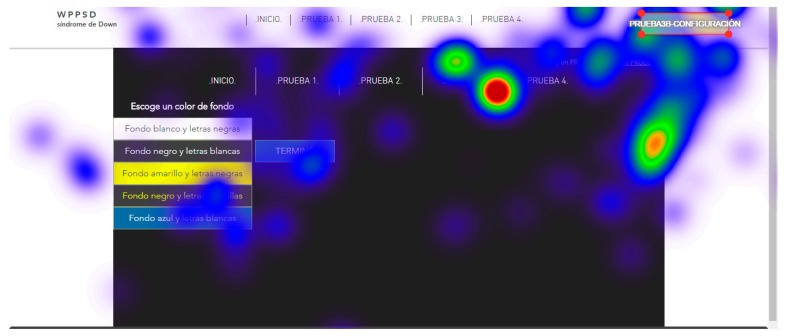
Contrast test black. Test 35.

**Figure 30 sensors-18-04047-f030:**
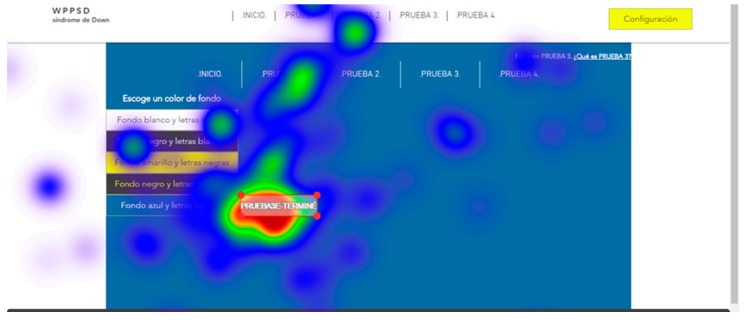
Blue contrast test. Essay 36.

**Figure 31 sensors-18-04047-f031:**
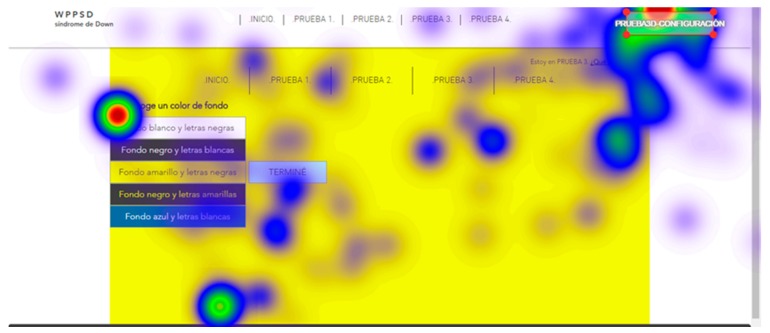
Yellow contrast test. Test 37.

**Figure 32 sensors-18-04047-f032:**
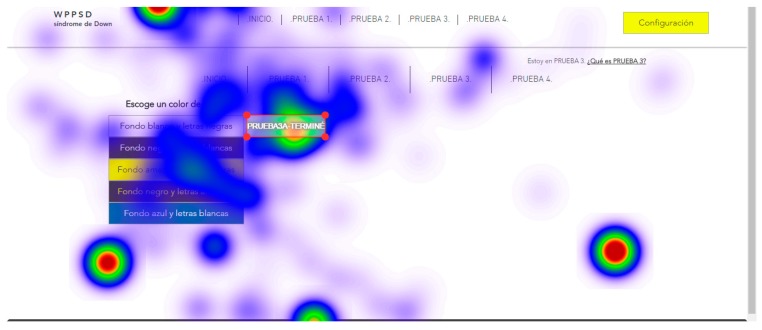
Temporal results of the test with black background and white letters. Comparison of the time spent in finding each of the elements. Test 35.

**Figure 33 sensors-18-04047-f033:**
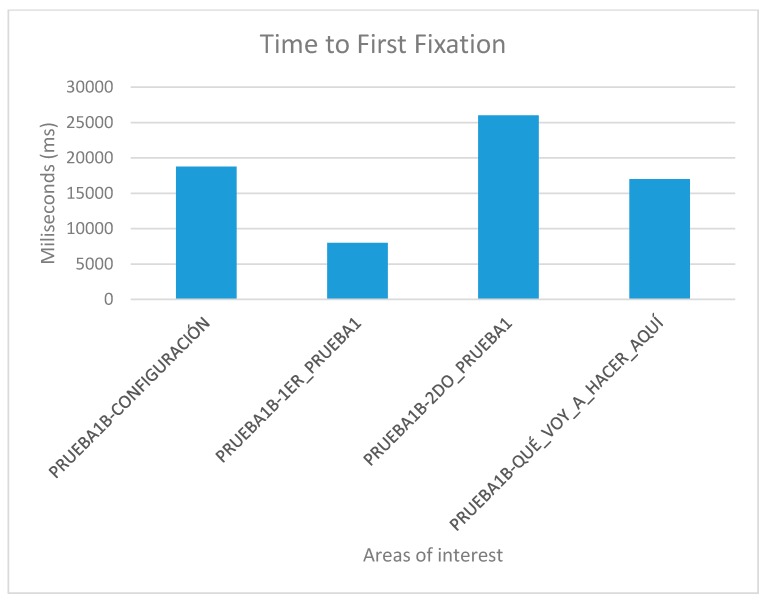
White Contrast test. Test 38.

**Figure 34 sensors-18-04047-f034:**
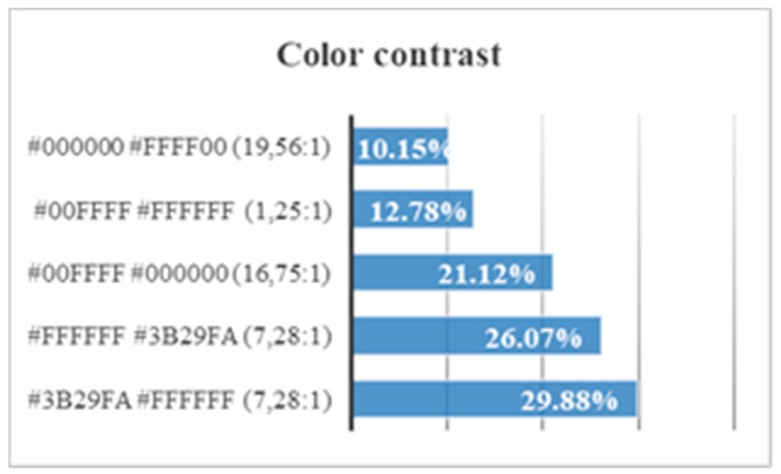
Evaluation 6.

**Figure 35 sensors-18-04047-f035:**
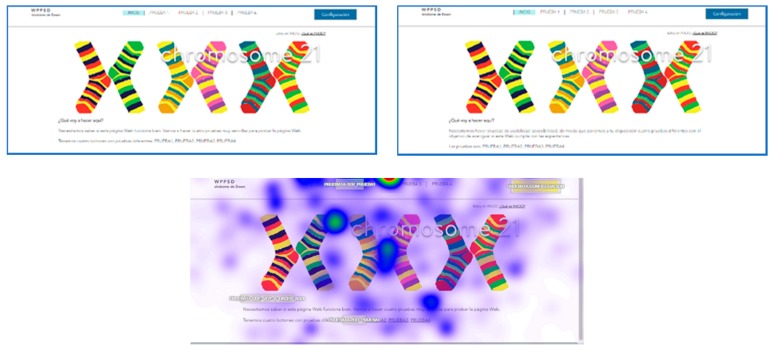
Links. Without links. Tracking buttons without links. Tests 39–41.

**Figure 36 sensors-18-04047-f036:**
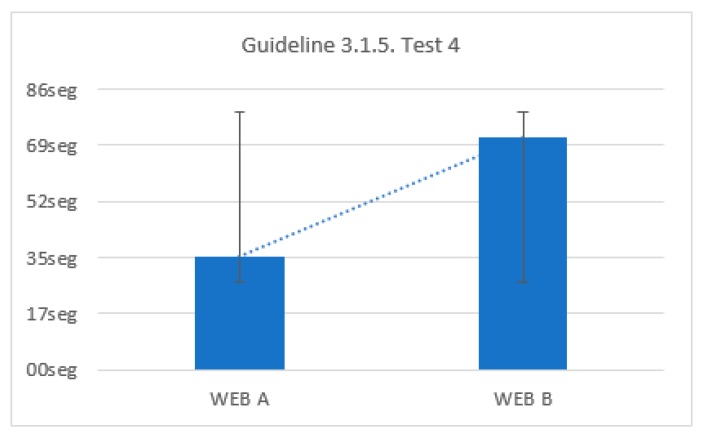
Links. Without links. Tests 39–40.

**Figure 37 sensors-18-04047-f037:**
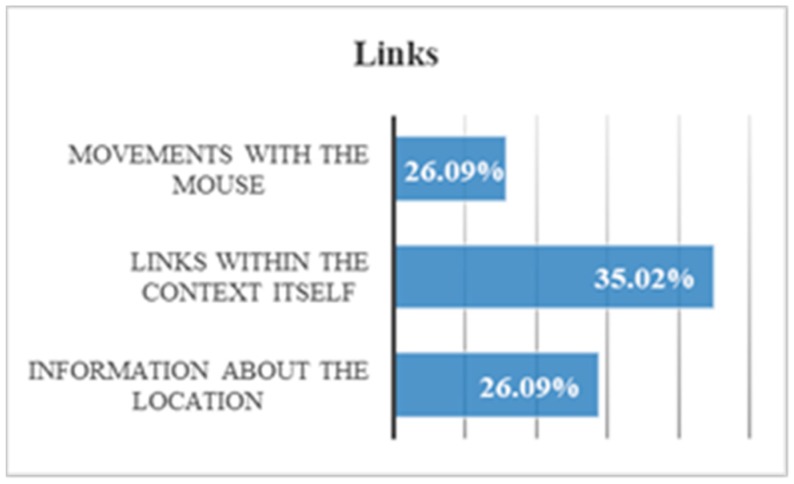
Evaluation 7.

**Figure 38 sensors-18-04047-f038:**
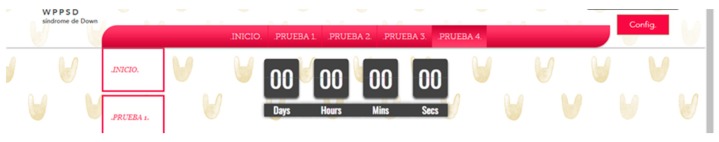
Visual temporal element. Test 43.

**Figure 39 sensors-18-04047-f039:**
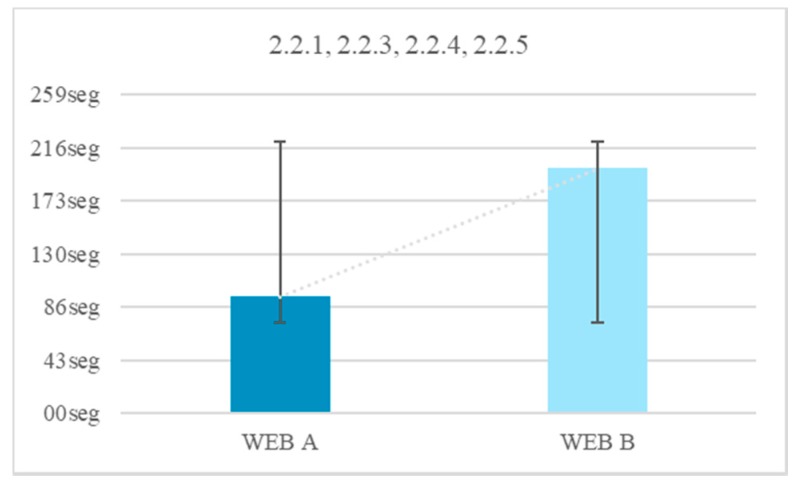
Without and with a temporal element. Test 43.

**Figure 40 sensors-18-04047-f040:**
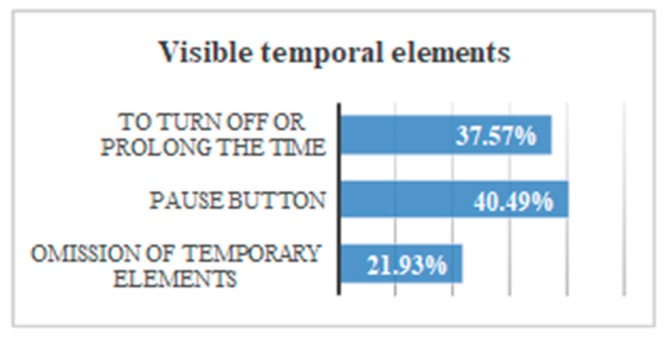
Evaluation 8.

**Figure 41 sensors-18-04047-f041:**
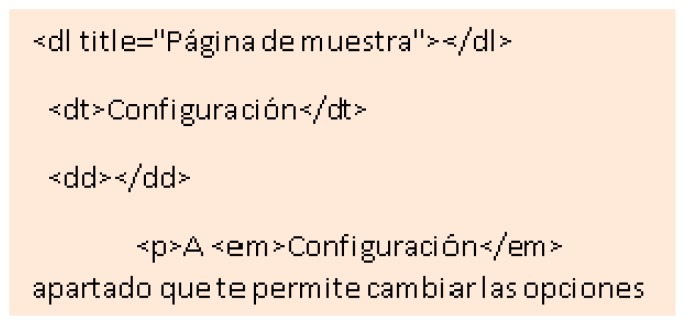
Textual content recommendation 3.

**Figure 42 sensors-18-04047-f042:**
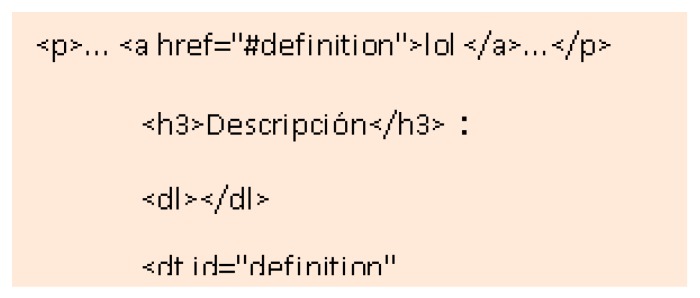
Textual content recommendation 5.

**Figure 43 sensors-18-04047-f043:**
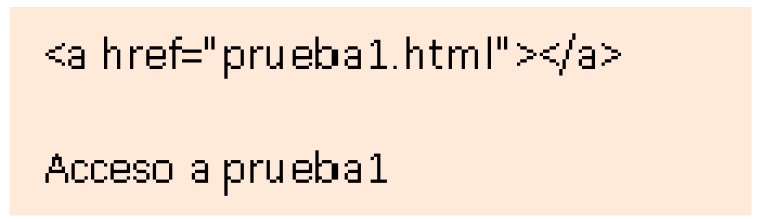
Link recommendation 1.

**Figure 44 sensors-18-04047-f044:**
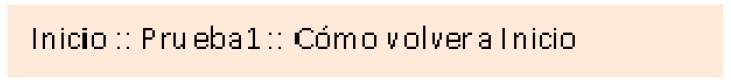
Link recommendation 3.

**Figure 45 sensors-18-04047-f045:**
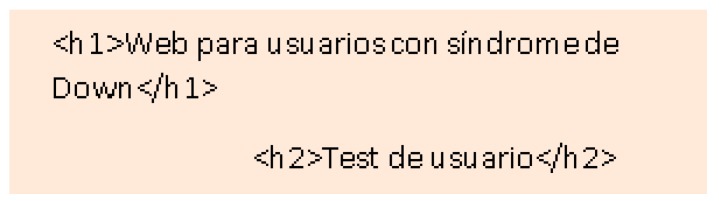
Labels recommendation 3.

**Figure 46 sensors-18-04047-f046:**
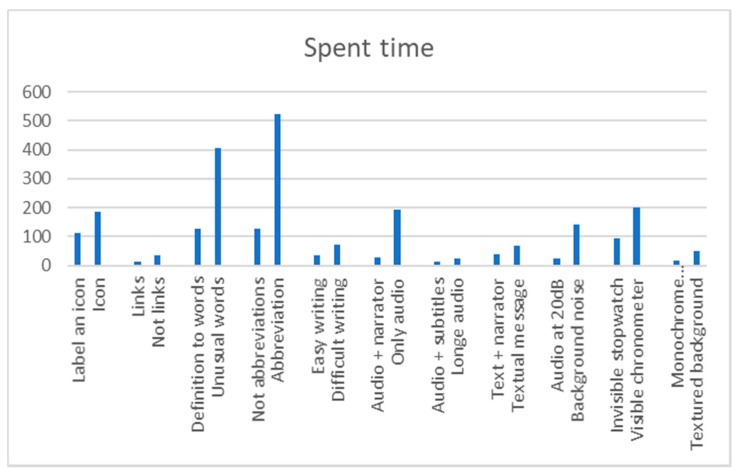
Spent time.

**Table 1 sensors-18-04047-t001:** Gender data.

**Gender**	**Man**	**Woman**	**Does Not Answer**
	46.43%	52.68%	0.89%

**Table 2 sensors-18-04047-t002:** Age data.

**Age**	**<14**	**15–34**	**35–54%**	**>55**	**Does Not Answer**
	2.54%	78.57%	17.03%	1.79%	0.07%

**Table 3 sensors-18-04047-t003:** Studies data.

**Studies**	**Does Not Know to Read Nor to Write**	**Primary Education**	**High School Studies**	**Vocational Studies**	**Training Studies**	**Other**	**Does Not Answer**
	2.68%	39.29%	34.82%	0.89%	4.46%	10.71%	7.14%

**Table 4 sensors-18-04047-t004:** Work data.

**Work**	**Yes**	**No**	**Does Not Answer**
	52.68%	47.32%	0.00%

**Table 5 sensors-18-04047-t005:** Age of participants in the user test and eye tracking test.

	Age
Average	18.0
Rank	14.0
Variance	17.7
Typical deviation	4.2

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
