# Peer review of "Web Page Design Recommendations for People with Down Syndrome Based on Users’ Experiences"

_sensors, 2018, doi:10.3390/s18114047_

Reviewer 1 Report

This paper described a survey result and the corresponding comprehensive set of tests on the computer usage behaviors of people who suffer from down syndrome. The concluded with a set of guideline on how to design web pages that are more conducive to people with down syndrome.

Before I can recommend to accept this paper, I hope the authors can improve the paper in the following way:

Conduct a more thorough comparison with related work. Even if no similar study on website design for down syndrome as claimed by the authors, I suppose there are rather similar study on website design for other form of disabilities. Such studies should be references and discussed. 

The presentation of the paper is very problematic. Several figures are not placed properly and hid some texts in the paper. Most figures are not very clear.

The paper used several acronyms that are never defined, such as WCAG, WAI. This is not acceptable for a scientific paper.

Author Response

Dear reviewer, first of all thank you for your comments. We appreciate your efforts very much by reviewing our work.

This paper described a survey result and the corresponding comprehensive set of tests on the computer usage behaviors of people who suffer from down syndrome. The concluded with a set of guideline on how to design web pages that are more conducive to people with down syndrome. Before I can recommend to accept this paper, I hope the authors can improve the paper in the following way:

Conduct a more thorough comparison with related work. Even if no similar study on website design for down syndrome as claimed by the authors, I suppose there are rather similar study on website design for other form of disabilities. Such studies should be references and discussed.

In this research, first an exhaustive analysis was carried out on the user profile and its visual, auditory, motor, communicative and cognitive deficits. Based on this, two studies were carried out:

1)     To demonstrate these theoretical claims, surveys were conducted. In surveys, it is asked about the general difficulties during web browsing and about specific web pages. The surveys were directed to specific users, they were not answered by any family member or assistant. The objective was to determine the problems that the users themselves believe they have and compare these data with the theoretical framework

2)     Each difficulty was related to a pattern of accessibility of the Accessibility Guidelines for Web Content.

It follows that WCAG web accessibility guidelines cannot be determined solely based on theoretical information. According to the theory, 90% of the existing recommendations are needed. However, after the surveys it is concluded that only 60% of the existing recommendations are useful.

From these data arises the need to design accessibility guidelines for this particular user profile and check whether these recommendations are really useful. Our conclusion is that a detailed analysis of a profile is necessary to determine any recommendation, and not based on theoretical observations. And then prove it.

The disabilities are so different that they demand different needs. We even found differences within the same profile, for this reason we filtered the participants. In order to have concrete results that help a focused group.

These particular needs vary so much in each user profile that what is concluded from some investigations is not always useful for others.

The presentation of the paper is very problematic. Several figures are not placed properly and hid some texts in the paper. Most figures are not very clear

Thank you again for your comments. We have reviewed and modified all mistakes found.

The paper used several acronyms that are never defined, such as WCAG, WAI. This is not acceptable for a scientific paper.

Finally, we have added the meaning of each acronym. We have modified it.

Reviewer 2 Report

I like the idea of this paper but found it VERY difficult to read due to the very poor written English and use of grammar. If the paper is to go forward it will need significant editing by a professional editor.

I think the authors needs to get some of the web accessibilty terminology correct - for example on line 133 the authors use the term ' (success guideline)' when I think they mean 'success criteria'.

In terms of the survey I really did not get the relevance beyond stating that 'a lot of people with Down Syndrome use the internet for various reasons' - a majority of the paper was spent on the user tests, so if the only role of the survey was to establish that people with Down Syndrome use the internet then I am not convinced it was needed.  I do appreciate that accessibility principles were applied in the design of the survey - though this brings me to quite a big concern.  The authors state that the survey and the user tests were designed around accessibility principles, but I could not find anywhere in the paper any discussion of the method they used to assess both the survey and user tests for accessibility conformance.  There are a number of automated assessment tools and manual assessment services that can test a site and its content and say 'these things meet accessibility guidelines A, AA or AAA' - I could find nothing about this.  It seems that the authors assumed that their accessible tools were indeed accessible. I find this to be a flawed assumption and would expect evidence that these tools WERE accessible and to what level of conformance (A or AA) to be included in the final version.

Finally, I thought the presentation of the statistical data was very basic - and I have never seen standard deviations referred to in the way these authors did - for example, the average age of a participant was 18 but the standard deviation was 24????  Why not just use tables of data, charts, diagrams and frequency distributions as one might find in a typical scientific paper?  Also, avoid the use of 'him' when generically referring to participants in the study - use 'him or her' or 'their'.

However, I found the conclusions interesting and compelling - it was just the journey to those conclusions that I think needs considerable polish. 

Author Response

Dear reviewer, first of all thank you for your comments. We appreciate your efforts very much by reviewing our work.

I like the idea of this paper but found it VERY difficult to read due to the very poor written English and use of grammar. If the paper is to go forward it will need significant editing by a professional editor.

Before submitting the paper, we send it to a professional translator. For this second revision, we have sent it again with the aim of improving our writing.

I think the authors needs to get some of the web accessibilty terminology correct - for example on line 133 the authors use the term ' (success guideline)' when I think they mean 'success criteria'.

Thanks for detecting this error. We have modified it. We also revise the rest of the paper trying to avoid these errors.

In terms of the survey I really did not get the relevance beyond stating that 'a lot of people with Down Syndrome use the internet for various reasons' - a majority of the paper was spent on the user tests, so if the only role of the survey was to establish that people with Down Syndrome use the internet then I am not convinced it was needed.  I do appreciate that accessibility principles were applied in the design of the survey - though this brings me to quite a big concern.

Thank you for this appropriate comment. The objective of the surveys was to determine if the number of recommendations useful for this profile, according to the theory, is adequate. It was determined that according to the theoretical references this user profile would need 90% of the existing guidelines. However, after the surveys, it is reduced to 60%.

In addition, the results of the surveys serve for the research tests. We have expanded this section with the aim of improving the understanding of it.

The authors state that the survey and the user tests were designed around accessibility principles, but I could not find anywhere in the paper any discussion of the method they used to assess both the survey and user tests for accessibility conformance. There are a number of automated assessment tools and manual assessment services that can test a site and its content and say 'these things meet accessibility guidelines A, AA or AAA' - I could find nothing about this. It seems that the authors assumed that their accessible tools were indeed accessible. I find this to be a flawed assumption and would expect evidence that these tools WERE accessible and to what level of conformance (A or AA) to be included in the final version.

Recommendations used in the survey have been extracted from the book Accessibility Handbook, by Katie Cunningham. Recommendations were also extracted from the WCAG about contrast, word spacing, etc. But the surveys were mainly solved in paper format, not online. And our main difficulty was the use of appropriate language. And an understandable wording to the interested user.

Advice from specialized personnel was taken. For example, they told us what kind of questions are solved more easily. And what number of answers are appropriate. It has been preferred to use the same communication scheme used in associations to avoid disorienting users when accessing information and answering forms. The review by the staff of the different associations has been very useful.

Since the survey was part of the investigation and we were not sure if the theoretical recommendations would be useful, trial and error tests were made with several participants. This helped to adjust the writing, until the questions could be answered autonomously by a specific profile.

Finally, I thought the presentation of the statistical data was very basic - and I have never seen standard deviations referred to in the way these authors did - for example, the average age of a participant was 18 but the standard deviation was 24????  Why not just use tables of data, charts, diagrams and frequency distributions as one might find in a typical scientific paper?  Also, avoid the use of 'him' when generically referring to participants in the study - use 'him or her' or 'their'.

However, I found the conclusions interesting and compelling - it was just the journey to those conclusions that I think needs considerable polish. 

Thanks to this suggestion we have detected a numerical error. It is already corrected. We have also included a table of data related to the ages of the participants. In this table the following data appear: average, rank, variance and typical deviation. We have changed some singular by plural to improve the comprehension of the writing.